

# 1 Fractal behavior of soil water storage at multiple depths

**Wenjun Ji[1], Mi Lin[1], Asim Biswas[1*], Bing C. Si[2], Henry W. Chau[3], and Hamish P. Cresswell[4]**
[1] Department of Natural Resource Sciences, McGill University, 21111 Lakeshore Road, Ste-Anne-de-
Bellevue, Quebec, Canada, H9X3V9
[2] Department of Soil Science, University of Saskatchewan, Saskatchewan, Canada, S7N5A8
[3] Department of Soil and Physical Sciences, Lincoln University, PO Box 85084, Lincoln, Christchurch,
New Zealand, 7647
[4] CSIRO Land and Water, Canberra, ACT, Australia, 2601
*Correspondence to*: A. Biswas (asim.biswas@mcgill.ca Phone: +1 514 398 7620; Fax: +1 514 398
10 7990)

**Abstract** Spatio-temporal behavior of soil water is essential to understand the science of
hydrodynamics. Data intensive measurement of surface soil water using remote sensing has
established that the spatial variability of soil water can be described using the principle of self-
similarity (scaling properties) or fractal theory. This information can be used in determining land
management practices provided the surface scaling properties hold at deep layer. Current study
examined the scaling properties of sub-surface soil water and its relationship to surface soil water,
thereby serving as the supporting information for the plant root and vadose zone models. Soil water
storage (SWS) down to 1.4 m depth at seven equal intervals was measured along a transect of 576
m for 5 years. The surface SWS showed multifractal nature only during the wet period (from
snowmelt until mid to late June with large SWS) indicating the need of multiple scaling indices in
transferring soil water variability information over multiple scales. However, with increasing
depth, the SWS became monofractal in nature indicating the need of single scaling index to
upscale/downscale soil water variability information. The dynamic nature made the surface layer
soil water in the wet period highly variable compared to the deep layers. In contrast, all soil layers
during the dry period (from late June to the end of the growing season with low SWS) were
monofractal in nature, probably resulting from the high evapotranspirative demand of the growing
vegetation that surpassed other effects. This strong similarity between the scaling properties at the
surface layer and deep layers provides the possibility of inferring about the whole profile soil water
dynamics using the scaling properties of the easy-to-measure surface SWS data.
**Keywords** Scaling, scale invariance, monofractal, multifractal, root zone, remote sensing



## 1 Introduction

Knowledge on the spatial distribution of soil water over a range of spatial scales and time has important hydrologic applications including assessment of land-atmosphere interactions (Sivapalan, 1992), performance of various engineered covers, monitoring soil water balance and validating various climatic and hydrological models (Rodriguez-Iturbe et al., 1995;Koster et al., 2004). However, high variability in soil is a major challenge in hydrology (Quinn, 2004) as the distribution of soil water in the landscape is controlled by various factors and processes operating at different intensities over a variety of scales (Entin et al., 2000). The individual and/or combined influence of these physical factors (e.g. topography, soil properties) and environmental processes (e.g. runoff, evapotranspiration, and snowmelt) gives rise to complex and nested effects, which in turn evolve a signature in the spatial organization (Western et al., 1999) or patterns in soil water as a function of spatial scale (Kachanoski and Dejong, 1988;Kim and Barros, 2002;Biswas and Si, 2011a). This complexity makes the management decision difficult at a scale other than the scale of measurement. Therefore, it is necessary to transfer variability information from one scale (e.g. pedon scale) to another (e.g. large catchment scale), which is called scaling.

The scaling of soil water is possible if the distribution of some statistical parameters (e.g., variance) remain similar at all studied scales. This feature, known as scale-invariance, means that the spatial feature in the distribution of soil water will not change if the length scales are multiplied by a common factor (Hu et al., 1997). Generally, the soil water will have a typical size or scale, a value around which individual measurements are centered. So the probability of measuring a particular value will vary inversely as a power of that value, which is known as the power law decay, a typical of scaling process. Now, as the spatial distribution of soil water follows the power law decay (Hu et al., 1997;Kim and Barros, 2002;Mascaro et al., 2010), the spatial variability can be investigated and characterized quantitatively over a large range of measurement scales using fractal theory (Mandelbrot, 1982). When the spatial distribution of soil water is the response of some linear processes, the scaling can be done using a single scaling coefficient over multiple scales and the distribution shows monofractal scaling behaviour. However, the spatial distribution of soil water is the nonlinear response of multiple factors and processes acting over a variety of scales and therefore needs multiple scaling indices (multifractal scaling) in quantifying spatial variability (Hu et al., 1997;Kim and Barros, 2002;Mascaro et al., 2010).



The multifractal scaling behaviour of soil water has been used in developing models to
downscale soil water estimate from remotely sensed measurements with a large foot print area.
The multifractal behaviour in the surface soil water as a result of temporal evolution of wetting
and drying has been reported from a sub-humid environment of Oklahoma by Kim and Barros
(2002). Mascaro et al. (2010) reported the multifractal behaviour of soil water, which was ascribed
as a signature of the rainfall spatial variability. Though these measurements can provide an
estimate of soil water over a large area quickly, they are limited to very few centimeters of the soil
profile. These studies reported the multifractal behaviour of only the surface soil water indicating
the superficial scaling properties. Surface soil layer is exposed to direct environmental forcing and
are most dynamic in nature. The scaling properties of surface soil water can be used for land
management practices provided the observed scaling properties holds for the deep layers such as
vadose zone or the whole soil profile. Understanding overall hydrological dynamics in soil profile
needs information on the scaling properties and the nature of the spatial variability of soil water
over a range of scales at deep layers as well (Biswas et al., 2012b). The information on the
similarity in the nature of the spatial variability of soil water between the surface layer and deep
layers may also help inferring about the soil profile hydrological dynamics. Therefore, the
objectives of this study were to examine the scaling properties of sub surface layers and their
relationship with surface layers at different initial soil water conditions over time. We have
examined the scaling properties of soil water storage at multiple depth layers and at soil layers
with increasing depth from the surface (cumulative depth) over a 5-year period from a hummocky
landscape from central Canada using the multifractal analysis. The relationship between the
scaling properties of the surface layer and the subsurface layers was also examined using the joint
multifractal analysis.
**2 Materials and Methods**
**2.1 Study site and data collection**
A field experiment was carried out at St. Denis National Wildlife Area (52°12′N lat. and 106°50′W
long.), which is located 40 km east of Saskatoon, Saskatchewan, Canada. The landscape of the
study area is hummocky with a complex sequence of slopes (10 to 15%) extending from different
sized rounded depressions to irregular complex knolls and knobs, a characteristic landscape of the
North American Prairie pothole region encompassing approximately 780,000 km$^2$ from north-





central United States to south-central Canada (National Wetlands Working Group, 1997). A
transect of 128 points (576 m long) extending in north-south direction was established in 2004 at
the study site to examine the soil water variation at field scale. The sample points were selected at
4.5 m regular interval along the transect to catch the systematic variability of soil water. Soil water
measurements were carried out at every 20 cm depth along the transect over the period of 2007 to
2011 and were used in this study to examine the fractal behavior of SWS at different depths of
over time. A detailed description of the study site, development of the transect, measurement of
soil water and the calibration of measurement instruments can be found in earlier publications from
this project (e.g. Biswas et al., 2012a).
**2.2 Data analysis**
Various methods including geostatistics, spectral analysis, and wavelet analysis have been used to
examine the scale-dependent spatial patterns of SWS. These methods generally deal with how the
second moment of SWS changes with scales or frequencies. When the statistical distribution of
SWS is normal, the second moment plus the average provide a complete description of the spatial
series. However, for other distributions (e.g. left skewed distribution), higher-order moments are
necessary for a complete description of the spatial series. For example, let's define the $q^{th}$ moment
of a spatial series $z$ as $z^q$. In this situation, for a positive value of $q$, the $q^{th}$ moment magnify the
effect of larger numbers and diminish the effect of smaller numbers in $z$. While, on the other hand,
for a negative value of $q$, the $q^{th}$ moment magnify the effect of small numbers and diminish the
effect of large numbers in the spatial series $z$. In this way, using variable moments, we can look at
the effect of the magnitude of the data in a series and characterize its spatial variability better.
There is a pressing need to summarize how these moments change with scales so that we can
compare and simulate spatially-variable SWS.
**2.2.1 Statistical self-similarity or scale invariance**
Soil water is highly variable in space and time. If the variability in the spatial/temporal distribution
remains statistically similar at all studied scales, the SWS is assumed to be self-similar (Evertsz
and Mandelbrot, 1992). Self-similarity, also called scale invariance, is closely associated with the
transfer of information from one scale to another (scaling). We used the multifractal analysis to
explore self-similarity or inherent differences in scaling properties of SWS in this study.



### 2.2.2 Multifractal analysis


On the spatial domain of the studied field, multifractal analysis was used to characterize the scaling
property of SWS by statistically measuring the mass distribution (Zeleke and Si, 2004). The spatial
domain or the data along the transect was successively divided into self-similar segments following
the rule of the binomial multiplicative cascade (Evertsz and Mandelbrot, 1992). This method
required that the two segments divided from a unit interval to be of equal length. With regards to
a unit mass $M$ (a normalized probability distribution of a variable or measured in a generalized
case) relating to the unit interval, the weight was also partitioned into $[h \times M]$ and $[(1-h) \times M]$,
where $h$ was a random variable ($0 \leq h \leq 1$) governed by a probability density function. Sequentially,
the new subsets with its associated mass were equally divided into smaller parts. In this way,
multifractal analysis was able to describe the scaling properties for the higher-order moments
compared to semivariogram which can only measure the scaling properties of the second moment.
In a special case, if the scaling properties do not change with $q$, the spatial series can be identified
as monofractal, when one scaling coefficient is enough to characterize. Generally, the multifractal
analysis is good at measuring the highly fluctuated mass (box size) as well as providing physical
insights at all scales regardless of any ad hoc parameterization or homogeneity assumptions
(Schertzer and Lovejoy, 1987).
For SWS spatial series, the scale-invariant mass exponent, was termed as $\tau(q)$ (Liu and Molz

(1997):

$$\langle [\Delta z(x)]^q \rangle \propto x^{\tau(q)} \qquad [1]$$
where $z$ was the SWS spatial series, $x$ was the lag distance and the symbol $\propto$ indicated
proportionality. The $\tau(q)$ is widely used in multifractal analysis. If the plot of $\tau(q)$ vs. $q$ [or $\tau(q)$
curve] has a single slope (i.e. a linear line), then the series is a simple scaling (monofractal) type.
If $\tau(q)$ curve is nonlinear and convex (facing downward), then the series is a multi-scaling
(multifractal) type. In this study, we used the UM model of Schertzer and Lovejoy (1987) to create
a linear reference line which represented the perfect monofractal type of scaling. Assuming the
conservation in mean value of SWS, this model simulated a cascade process with a scaling function
in an empirical moment. It is thus used here to compare and characterize the observed scaling
properties with a reference to the monofractal behavior. The goodness-of-fit between the $\tau(q)$
curves and the UM model was tested using the chi-square test. The sum of squared residuals



(SSRs) between the $\tau(q)$ curve and the UM model was also calculated to test the deviation. The
$\tau(q)$ curves over the range of $q$ values (in this study -15 to 15 at 0.5 interval) were fitted with a
linear regression line (referred to as a single fit). The linear fitting of the $\tau(q)$ curves with $q<0$ and
$q>0$ (referred to as segmented fit) were also completed. The difference between the mean of slopes
and segmented fits (for positive and negative $q$ values) was tested using the Student's $t$ test.
With similar manner to Eq. [1], the $q^{th}$ order normalized probability measure of SWS, $\mu(q,\varepsilon)$
(also known as the partition function), is proved to vary with the scale size, as below
$$\mu_i(q,\varepsilon)=\frac{[p_i(\varepsilon)]^q}{\sum_i [p_i(\varepsilon)]^q} \propto (\varepsilon/L)^{\tau(q)} \qquad [2]$$
where $\varepsilon$ is scale size in the $i^{th}$ segment and $p_i(\varepsilon)$ is the probability of a measure and measures the
concentration of a variable of interest (e.g. SWS) by dividing the value of the variable in the
segment to the whole support length (e.g. to the whole transect of length $L$ units) (Meneveau et al.,
1990;Evertsz and Mandelbrot, 1992). The mass exponent $\tau(q)$ was related to the probability of
mass distribution of SWS.
Moreover, the fractal dimension of the subsets of segments in scale size $\varepsilon$ was measured by the
multifractal spectrum $f(q)$. When a coarse Hölder exponent (local scaling indices) of $\alpha$ was in the
limit as $\varepsilon \to 0$, $f(q)$ was calculated as below (Evertsz and Mandelbrot, 1992):
$$f(q)=\lim_{\varepsilon\to 0}\left(log\left(\frac{\varepsilon}{L}\right)\right)^{-1}\sum_i \mu_i(q,\varepsilon) log\, \mu_i(q,\varepsilon) \qquad [3]$$
and the local scaling indices, $\alpha$, were given by
$$\alpha(q)=\lim_{\varepsilon\to 0}\left(log\left(\frac{\varepsilon}{L}\right)\right)^{-1}\sum_i \mu_i(q,\varepsilon) log\, p_i(\varepsilon) \qquad [4]$$
Noting that $f(\alpha)$ was determined through the Legendre transform of the $\tau(q)$ curve:
$f(\alpha)=q\alpha(q)-\tau(q)$ (Chhabra and Jensen, 1989).
The multifractal spectrum is a powerful tool in portraying the similarity and/or differences
between the scaling properties of the measures (e.g. SWS). This spectrum also enabled us to
examine the local scaling property. The width of the spectrum ($\alpha_{max}$ - $\alpha_{min}$) was used to examine



the heterogeneity in the local scaling indices. The wider the spectrum, the higher was the
heterogeneity in the distribution of SWS and vice versa. Similarly, the height of the spectrum
corresponded to the dimension of the scaling indices. The small $f(q)$ values indicated rare events
(extreme values in the distribution), whereas the largest value was the capacity dimension ($D_0$)
obtained at $q = 0$.

In addition to the multifractal spectrum, [$f(q)$ vs. $\alpha(q)$], for many practical applications, we

used models to incorporate a few selected indicators to describe the scaling property and variability
of a process. One of the widely used models for multifractal measure were the generalized
dimensions, which was calculated as below:
$$D_q = \frac{1}{q-1} \lim_{\varepsilon \to 0} \frac{log \sum_i p_i(\varepsilon)}{log(\varepsilon)}$$    [5]
when $q = 1$, $D_1$ was referred to as the information dimension (also known as entropy dimension)
which provided information about the degree of heterogeneity in the measure distribution in
analogy to the entropy of an open system in thermodynamics (Voss, 1988). If the value of $D_1$ is
close to unity, it indicated the evenness of measures over the sets of cell size, while the value
approaching 0 indicated a subset of scale in which the irregularities were concentrated. The $D_2$,
known as the correlation dimension, was associated with the correlation function and measured
the average distribution density of the SWS (Grassberger and Procaccia, 1983). For a monofractal
distribution, the $D_1$ and $D_2$ tend to be equal to the $D_0$. The same value of $D_0$, $D_1$ and $D_2$ indicates
that the distribution exhibits perfect self-similarity and is homogeneous in nature. Contrarily, in
multifractal type scaling, the $D_1$ and $D_2$ tend to be smaller than $D_0$, showing $D_0 > D_1 > D_2$.
Accordingly, the $D_1/D_0$ value can be used to describe the heterogeneity in the distribution
(Montero, 2005). The value equal to 1 indicated exact mono-scaling of the distribution.
**2.2.3 Joint multifractal analysis**
While the multifractal analysis characterized the distribution of a SWS spatial series along its
geometric support, the joint multifractal analysis was used to characterize the joint distribution of
two SWS spatial series along a common geometric support. As an extension of the multifractal
analysis, the length of the datasets was also divided into several segments in size ε. Two variables
($P_i(\varepsilon)$ and $R_i(\varepsilon)$ representing two spatial series of SWS) were used here to measure the probability





of the measure in the $i$[th] segment, when $P_i(\varepsilon) \infty (\varepsilon/L)^\alpha$ and $R_i(\varepsilon) \infty (\varepsilon/L)^\beta$. Among them, $\alpha$ and $\beta$
were the local singularity strength which respectively represented the mean local exponents of
$P_i(\varepsilon)$ and $R_i(\varepsilon)$ in the corresponding expressions above. The partition function for the joint
distribution of $P_i(\varepsilon)$ and $R_i(\varepsilon)$, was calculated as below (Chhabra and Jensen, 1989; Meneveau et
al., 1990; Zeleke and Si, 2004):
$$\mu_i(q,t,\varepsilon) = \frac{p_i(\varepsilon)^q \cdot r_i(\varepsilon)^t}{\sum_{j=1}^{N(\varepsilon)} \left[ p_j(\varepsilon)^q \cdot r_j(\varepsilon)^t \right]}.$$    [6]
where the normalized $\mu$ is partition function, $q$ and $t$ were the real numbers for weighting. And the
aforementioned local singularity strength (coarse Hölder exponents) $\alpha$ and $\beta$ were the function to
$q$ and $t$ as well:
$$\alpha(q,t) = -\left[\ln(N(\varepsilon))\right]^{-1} \sum_{i=1}^{N(\varepsilon)} \left[\mu_i(q,t,\varepsilon) \cdot \ln(p_i(\varepsilon))\right]$$    [7]
$$\beta(q,t) = -\left[\ln(N(\varepsilon))\right]^{-1} \sum_{i=1}^{N(\varepsilon)} \left[\mu_i(q,t,\varepsilon) \cdot \ln(r_i(\varepsilon))\right].$$    [8]
To indicate the dimension of the joint distribution, the multifractal spectra $f(\alpha,\beta)$, was given by
$$f(\alpha,\beta) = -\left[\ln(N(\varepsilon))\right]^{-1} \sum_{i=1}^{N(\varepsilon)} \left[\mu_i(q,t,\varepsilon) \cdot \ln(\mu_i(q,t,\varepsilon))\right].$$    [9]
In fact, the joint partition function in Eq. [6] can be simplified to Eq. [2] when $q$ or $t$ is equal to 0.
In this case, the joint multifractal spectrum was transformed to the multifractal spectrum with a
single measure. When both value of $q$ and $t$ were 0, $f(\alpha,\beta)$ reached maximum and indicated box
dimension of the geometric support of the measures. Pair value of $\alpha$ and $\beta$ were determined by
variable $q$ and $t$. The Pearson correlation coefficient was used to quantitatively describe their
relations across similar moment orders. In addition, correlation coefficients between the surface
layer and subsurface layers were used as well to examine the similarity in the scaling properties.
**3 Results**
**3.1 Spatial pattern of soil water storage at different depths**



Average SWS for the surface 0-20 cm layer over five year period was 5.51 cm. A slight decrease
in SWS was observed at the immediate deep layer (20-40 cm) and a gradual increase thereafter.
Five-year average SWS was 5.45 cm, 5.48 cm, 5.56 cm, 5.61 cm, 5.69 cm and 5.77 cm for the 20-
40 cm, 40-60 cm, 60-80 cm, 80-100 cm, 100-120 cm and 120-140 cm layers, respectively (Table
1). Average SWS for a single measurement varied from 3.40 cm to 7.16 cm. The highest average
SWS was observed on 29 June 2011. The study area received large amount of spring rainfall during
2011 leading to the high SWS in the surface layer. The lowest average SWS was observed on 23
August 2008, which was one of the driest summer within the five-year study period. The highest
average SWS (on 29 June 2011) at the surface layer gradually decreased to 6.55 cm and the lowest
average SWS (on 23 August 2008) at the surface layer gradually increased to 5.28 cm at the 120-
140 cm layer (Table 1). This yielded a bigger range (3.76 cm) in the average SWS at the surface
layer compared to that at the deepest layer (1.27 cm). A big range (2.00 cm) in the standard
deviation (maximum=2.43 cm and minimum=0.43 cm) of the measurement at the surface layer (0-
20 cm) was also observed compared to that at the deepest layer (120-140 cm; maximum=1.28 and
minimum=0.76). This indicated large variations in SWS at the surface layer and gradually
decreased at deeper layers. The coefficient of variations (CVs) at the surface layer (0-20 cm) varied
from 10% to 43% and the deepest layer (120-140 cm) varied from 13% to 23% (Supplementary
Table S.1).
The maximum SWS at the surface layer also varied widely (maximum=13.96 cm and
minimum=4.64 cm) compared to the deepest layer (maximum=9.81 cm and minimum=6.72 cm)
(Table 1). There was a gradual decrease in the maximum value and increase in the minimum value
from the surface to the deepest layer. A similar trend was also observed for the minimum SWS at
different layers. The maximum SWS at different layers was much localized. For example, there
was high SWS at different layers at the locations of 100 to 140 m and 225 to 250 m from the origin
of the transect. These locations had very high SWS compared to the field-average and were situated
in the depressions while low SWS was observed on the knolls.
The variations in SWS with time were evaluated within a year. There was little change in the
average SWS over measurements within the years from 2007-2011 except 2008 (Table 1). For
example, average SWS was 6.47 cm, 6.03 cm, 6.54 cm, and 6.33 cm on 6 April 2010, 19 May
2010, 14 June 2010 and 28 September 2010, respectively. However, the average SWS in 2008
drops from 6.28 cm on 2 May 2008 to 3.51 cm on 17 September 2008 in the surface 0-20 cm layer.

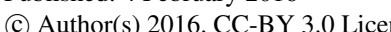



This falling trend was even observed at all soil layers. When compared between years, the trend
over time and with depth was very similar in 2007 and 2009 while slightly different between 2010
and 2011 (Table 1). A decreasing trend of the variability was also observed with time. For example,
the CV of the surface layer was around 28% on 2 May 2008, which gradually decrease to around
13% on 17 September 2008 (Supplementary Table S.1).
The average water storage for soil layers with increasing depth was also calculated by adding
the individual layers together. The time-averaged values of SWS were 10.96 cm, 16.44 cm, 22.00
cm, 27.61 cm, 33.30 cm and 39.07 cm for the 0-40 cm, 0-60 cm, 0-80 cm, 0-100 cm, 0-120 cm
and 0-140 cm, respectively (Supplementary Table S.2). The CV of the 0-20 cm layer was the
highest during the wet period and gradually declined to the smallest during the dry period
(Supplementary Table S.3). The variability also gradually increased with depth. This trend with
depth and time has also been verified by the standard deviation of measurement.
**3.2 Statistical scale invariance**
The distribution of a statistical measure is considered as fractal (monofractal/multifractal) provided
the moments obey the power law (Evertsz and Mandelbrot, 1992). The power law relationships
and the statistical scale invariance were evaluated using a log-log plot of the aggregated variance
of SWS spatial series at different depths of soil layers and the level of disaggregation (or scales)
at different $q$ values or statistical moments. The linear relationship of the logarithm of the variance
with scale indicated the presence of statistical scale invariance (Fig. 1). The scale invariance was
observed for all measurements and at all depths though only all depths of selected three
measurements were presented as example. The coefficient of determination ($r^2$) for a linear fit
($n$=7) was between 0.99 and 1.00 (significant at $P$=0.001) for any measurement days and depths.
The scale invariance was also observed for SWS at soil layers with cumulative depths.
**3.3 Multifractal analysis**
The $\tau(q)$ curves for the surface layer displayed deviation from the UM model during the wet period
(Fig. 2). A high SSR value was observed between the $\tau(q)$ curves and the UM model. Nonlinearity
in the $\tau(q)$ curve was observed and the slopes of the segmented fit of the $\tau(q)$ curves were
significantly different from each other. For example, the SSR values between the $\tau(q)$ curve and
the UM model were 27.74 and 50.49 for the surface layer (0-20 cm) on 2 May 2008 and 31 May
2008, respectively. The slopes of the $\tau(q)$ curve for (single fit) were 0.97 and 0.96, respectively for



the surface layer of 2 May 2008 and 31 May 2008 (Fig. 2). The slopes of the segmented fit for
these measurements were 1.04 ($q<0$) and 0.87 ($q>0$) and, 1.06 ($q<0$) and 0.82 ($q>0$), respectively
(Fig. 2; Supplementary Table S.4).
With the maximum deviation at the surface layer, the $\tau(q)$ curves gradually became very similar
to the UM model with depth. The SSR value decreased considerably in the deep layers. The slopes
of the $\tau(q)$ curve (single fit) became almost unity with no significant difference with the UM model.
There was no significant difference between the slopes of the segmented fit. For example, the SSR
value was 6.17, 4.98, 8.80, 8.50, 8.86, and 6.16 respectively for the 20-40, 40-60, 60-80, 80-100,
100-120, and 120-140 cm layer of 2 May 2008 (Supplementary Table S.4). The slopes (single fit)
for these layers were 0.99, 1.00, 1.01, 1.01, 1.00, and 0.99, respectively (Fig. 2). The slopes of the
segmented fit were also very close to unity with no significant difference between them.
The SSR values gradually decreased and the slopes became almost unity with the increase of
depth of soil layers (Fig. 3). For example, the SSR values were 14.11, 9.31, 7.71, 6.86, 6.71 and
6.30 and the slopes (single fit) were 0.98, 0.99, 0.99, 1.00, 1.00, and 1.00, respectively for 0-40,
0-60, 0-80, 0-100, 0-120 and 0-140 cm layer (Supplementary Table S.5). The slopes of the
segmented fit for the $\tau(q)$ curve became almost the same as soil layers going deeper (Fig. 3). The
linearity of the $\tau(q)$ curves was gradually strengthened and the SSR value gradually fell with the
depth increase of soil layers at any time. A statically significant difference was observed between
the slopes of the $\tau(q)$ curves in segmented fitting at the surface layer of first three measurements
in 2007 (Supplementary Fig. S.1), two measurements in 2008 (Fig. 3), three measurements in 2009
(Supplementary Fig. S.2), and all measurements in 2010 and 2011 (Fig. 3).
A decreasing trend in the SSR value was also observed over time within a year. During the dry
period, the slopes (single fit and segmented fit) became almost unity with no significant difference
(Supplementary Table S.6). For example, the SSR value was 14.12, 8.25, 1.30, 1.46, and 0.52 and
the slope was 0.99, 0.99, 1.00, 1.00, and 1.00, respectively for the surface layer (0-20 cm) of 21
June 2008, 16 July 2008, 23 August 2008, 17 September 2008 and 22 October 2008 (Fig. 2).
Similarly, a small SSR value and consistent slope were also observed at the deepest layer (120-
140 cm). The SSR values of the 120-140 cm were 2.47, 2.47, 3.31, 3.44 and 4.57, respectively for
the measurements on 21 June 2008, 16 July 2008, 23 August 2008, 17 September 2008 and 22





October 2008 (Supplementary Table S.6). The slope (single fit) for all these measurements was
equal to 1.01 (Fig. 2). There was very little difference in the slopes of the segmented fits.
A significant difference in the slopes of the segmented fit was observed for the surface layer
(0-20 cm) of three measurements in 2007 (17 July, 7 August, and 1 September; Supplementary
Fig. S.1), and three measurements in 2009 (21 April, 7 May, and 27 May) (Supplementary Table
S.4; Supplementary Fig. S.2). The trend in deep layers over time was very similar to that of 2008.
However, the trend in the SSR values and the slopes with time was scarcely different between
2010 and 2011 (Supplementary Table S6). There was very little difference in the SSR values at
different time of the year. For example, the SSR value for the surface layer (0-20 cm) was 20.79,
27.18, 24.63 and 26.66 and the slope (single fit) was 0.97, 0.97, 0.97, and 0.97, respectively for
the measurements on 6 April 2010, 19 May 2010, 14 June 2010, and 28 September 2010 (Fig. 2).
The slope of the segmented fit of the surface layer (0-20 cm) was statistically significant for all
measurements in 2010 and 2011 (Fig. 2). However, the trend with depth was similar to other years
(Supplementary Table S.7).
The height of the multifractal spectrum at different depths of measurement over time was very
similar. The width of the spectrum ($\alpha_{max}$-$\alpha_{min}$) varied with depth and time. Generally, a comparative
large value of $\alpha_{max}$-$\alpha_{min}$ was observed at the surface layer during the wet period and the value
gradually became smaller at depths. For example, the value of $\alpha_{max}$-$\alpha_{min}$ for the surface soil layer
(0-20 cm) was 0.23 and 0.31, respectively for the measurements of 2 May 2008 and 31 May 2008.
Meanwhile, the value of $\alpha_{max}$-$\alpha_{min}$ for the soil layers of 20-140 cm with 20 cm increment was 0.15,
0.14, 0.19, 0.20, 0.20, and 0.18 for 2 May 2008 and 0.25, 0.19, 0.11, 0.14, 0.12, and 0.11 for 31
May 2008, respectively (Fig. 4). In the later part of the year, the width of the spectrum gradually
decreased (Supplementary Table S.8). For example, the $\alpha_{max}$-$\alpha_{min}$ values were 0.19, 0.16, 0.07,
0.08, and 0.05, respectively for the surface layer measurement of 21 June 2008, 16 July 2008, 23
August 2008, 17 September 2008 and 22 October 2008. Similar trend in values of $\alpha_{max}$-$\alpha_{min}$ was
also observed at deep layers (Fig. 4).
The trend of the $\alpha_{max}$-$\alpha_{min}$ values in 2007 and 2009 was very similar to that of 2008
(Supplementary Table S.8). A higher value of $\alpha_{max}$-$\alpha_{min}$ was observed in first three measurements
of 2007 (Supplementary Fig. S.5) and three measurements of 2009 (Supplementary Fig. S.6).
However, the values in the surface layer (0-20 cm) of measurements in 2010 and 2011 were always



higher compared to the deep layers (Fig. 4). There was no decreasing trend in values for the surface
layer over time. For example, the $\alpha_{max}$-$\alpha_{min}$ value was 0.21, 0.24, 0.21, and 0.22, respectively for
the measurements on 6 April 2010, 19 May 2010, 14 June 2010, and 28 September 2010 (Fig. 4).
However, the trend in the $\alpha_{max}$-$\alpha_{min}$ value of deep layers was similar to that of other years. A similar
trend was observed for cumulative SWS with increasing depth over the years (Fig. 5). Generally,
the value of $\alpha_{max}$-$\alpha_{min}$ was also small with the highest in the 0-20 soil layers and gradually
decreased with depth (Fig. 5; Supplementary Table S.9).
Generally, the $D_1$ and $D_2$ values for different depths of different measurements were very close
to 1 (only varied at 3 decimal points; Supplementary Table S.10). Specifically, the $D$ values for
the surface layer during the wet period increased at high $q$ values. For example, the first three
measurements in 2007 and 2009 all presented high $D$ values at high $q$ values (Supplementary Figs.
S.9 and S.10). This high $D$ value gradually decreased in the dry period of the year. For example,
the $D$ value with positive $q$ was high in the surface layer of 2 May 2008 and 31 May 2008 (Fig.
6), whereas it gradually decreased at the later part of the year (e.g. 17 September 2008).The trend
with time and depth in 2007 and 2009 was very similar to that of 2008 (Supplementary Tables
S.10 and S.11). A consistent high $D$ value was observed in the surface layer for all 2010 and 2011
measurements (Fig. 6). The trend in $D$ values with depth in 2010 and 2011 was also similar to
other years. A high value of $D_1$ and $D_2$ were also observed at all layers of cumulative depths for
all measurements (Fig. 7; Supplementary Table S.11).
**3.4 Joint multifractal analysis**
There were strong correlations between the scaling property of the joint distribution of the surface
soil layer and the deep soil layers. The correlation between the surface 0-20 cm and the deep layers
on 2 May 2008 (wet period) was larger than 0.9 (significant at $P$=0.001; Table 2). The highest
correlation was observed between the layers closest to each other. The correlations gradually
increased over time and showed high consistency between different layers on 17 September 2008
(Table 2). A very similar trend was observed in other years.
**4 Discussion**
The amount of water stored in soil layers is the result of the dominant underlying hydrological
processes. Located in semi-arid climate, the study area receives about 30% of the long term annual
average precipitation as snowfall during winter months (Pomeroy et al., 2007). Generally, the



depressions receive snow from surrounding uplands or knolls as redistributed by strong prairie
wind (Pomeroy and Gray, 1995;Fang and Pomeroy, 2009). The snow melts within short period of
time during the early spring and contributed a large amount of water. The frozen ground restricts
infiltration and redistributes excess water within the landscape with greater accumulation in
depressions (Fig. 8) (Gray et al., 1985). Apart from the snowmelt, the spring rainfall also
contributes to the water inflow in the landscape (Fig. 8). This created a spatial pattern of SWS that
was almost a mirror image of the spatial distribution of relative elevation (Biswas and Si, 2011a,
b;Biswas et al., 2012a).
In the spring, the sources of water loss were the deep drainage and the evaporation. . As the
loss of water through deep drainage in the study area was as low as 2 to 40 mm per year, occurring
mainly through the fractures and preferential flow paths (Hayashi et al., 1998;van der Kamp et al.,
2003), the major loss occurred mainly through evaporation from the surface of the bare ground
and standing water in depressions. These processes lose a very small amount of water compared
to the input of the water in spring and early summer leaving the soil wet. Moreover, the surface
soil with high organic matter content and low bulk density stored larger amount of water than the
deep layers where the organic matter gradually decreased and the bulk density increased.
Reflecting the long-term history of vegetation growth in the landscape, the variability of organic
matter content (CV=41%) may be one of the main factor of the high variability in surface layer
SWS (Biswas and Si, 2011c)..
As the vegetation developed in summer, strong evapotranspiration resulted in the lowest
average SWS in a year. High amount of water in the depressions allowed grasses to grow faster
and transpire more water comparing to the knolls (Fig. 8). For example, the aquatic vegetation
growth within the depressions was as high as 2 m, while the grasses on the knolls grew to a
maximum up to a meter tall. The uneven growth of vegetation and the high evapotranspirative
demand in summer narrowed the range of SWS. Stronger demand extracted more water from the
soil where available and comparative less water from the soil where the availability was restricted,
thus reducing the disparities between maximum and minimum values. This variable water uptake
was visible in the growth of vegetation in the later part of the growing season as well (Fig. 8). The
reduction in the range of SWS was the largest in the surface layer and gradually decreased at deep
layers. This is because the surface layer was exposed to various environmental forcing and was
very dynamic in nature.  For example, plants can take up more than 70% of the water they need



from the top 50% of the root zone (Feddes et al., 1978). This dynamic behavior of the surface layer
exhausted readily available water and finally reduced the range in water storage. This decrease in
range also happened in the later part of the growing season.
The multifractal and joint multifractal analyses explained the scaling behavior of SWS at
different depths over time. The linearity in the log-log plot between the aggregated variance in
SWS and the scale at all soil layers over time indicated the presence of scaling laws (Fig. 1). The
mass exponent, $\tau$ calculated over a range of moment orders ($q$) was used to examine the scaling
behavior (monofractal and multifractal). The shape of the curve described the type of scaling
involved. The curve with a single slope implied a monofractal scaling, while a convex downward
curve with different slopes for negative and positive moment orders implied a multiple scaling
(multifractal) (Evertsz and Mandelbrot, 1992). The deviation in the scaling property of SWS from
the monofractal was also examined by comparing the $\tau(q)$ curve with the theoretical UM model
and the SSR between them (Fig. 2). The near unity slope of the $\tau(q)$ curves and the insignificant
difference from the UM model indicated a monofractal type scaling at all layers except the surface
layer during the wet period (until mid to late June) where a multifractal behavior led to a slight
convex downward curve (Fig. 2). This was also supported by a significant difference between the
slope of single and segmented fit in the surface layer during the wet period.
Generally during the wet period, excess water fills and drains macropores quickly and creates
variations in SWS. Variations in the evaporation due to uneven solar incidence over micro-
topography also triggered SWS variability in the surface layer. Additionally, the snow melt and
the release of water controlled by local (e.g. soil texture) and non-local (e.g. topography) factors
also affected the spatial distribution of SWS, making it more heterogeneous in the wet period
(Grayson et al., 1997; Biswas and Si, 2012). Contrarily, as depth increased, less impact of
environmental forcing tended to create less variability in SWS and exhibited monofractal behavior
which was consistent with the uniform slope shown in Figure 2. During the dry period or later part
of the growing season, the SWS storage variability at all depths was small and exhibited
monofractal behavior (Fig. 2). Accordingly, the deeper layers in the wet period and all layers in
the dry period can be accurately represented by only one scaling exponent while the surface layer
in the wet period may require a hierarchy of exponents to describe scaling property. A similar trend
was observed in SWS of cumulative depth layers (Fig. 3). Resulting from increasingly buffering





capacity of the deeper soil layers, the variability of cumulative SWS overlaid the multifractal
nature of the surface layer, and finally exhibited monofractal behavior in general.

The scaling patterns of SWS at different depths and different periods were further examined
using multifractal spectrum [$f(q)$ vs. $\alpha(q)$] (Fig. 4 & Fig. 5). The degree of convexity was used to
characterize the heterogeneity of scaling exponents or the degree of multifractality. Large value of
$\alpha_{max}$-$\alpha_{min}$ indicated stronger heterogeneity in the local scaling indices of SWS or cumulative SWS
and vice versa. The largest value for the surface layer(s) in the wet period indicated the most
multifractal behavior of SWS. However, the value decreased with depth and gradually converged
in deep layers (Fig. 4). This decline manifested a conformity in the scaling behavior of SWS at
deeper layers. Over time, the $\alpha_{max}$-$\alpha_{min}$ value of the surface soil layer decreased and became very
similar to that of deep layers. This indicated a reduction in the degree of multifractality for surface
soil layers from the wet period to the dry period. A consistent $\alpha_{max}$-$\alpha_{min}$ value for all depths during
the dry period suggested the homogeneity and least multifractal nature of SWS. A similar behavior
was observed in the cumulative SWS (Fig. 5).

To sum up, both the unity slope of the $\tau(q)$ curves (Fig. 2 and Fig. 3) and the degree of
convexity of the $f(q)$ spectrum (Fig, 4 & Fig. 5) jointly demonstrated that dynamic behavior of
surface soil layers in the wet period made SWS highly variable and exhibited multifractal nature,
while less environmental forcing and increased buffering capacity of deep layers led to
monofractal nature. As a result, multiple scaling exponents were required to characterize the
variability of SWS in the surface layer during the wet period, while less number of exponents was
necessary for deeper layers during wet period or all layers during dry period.

The height of the spectrum, $f(q)$ revealed the dimension or frequency distribution of the scaling
indices. A low height of $f(q)$ curve indicated rare events or extreme values in the distribution, while
a high value represented uniform distribution in all segments. A very similar height of the $f(q)$
curve for all depths and all periods indicated a consistent frequency distribution of the scaling
indices. Additionally, the position and the symmetry of the curve revealed the distribution of
scaling exponents. A symmetric $f(q)$ curve indicated uniform distribution of the scaling exponents.
The left side of the spectrum corresponded to the large SWS that were amplified by the positive
values of $q$ while the right side indicated smaller SWS that were amplified by negative $q$ values.





Surface one or two layers during the wet period tended to exhibit longer tail of the curve on
the left, showing more heterogeneity in the distribution of large values. However, when stepping
into the dry period, the spectrum tended to display a longer tail on the right compared to the left
side, suggesting more heterogeneity in the distribution of smaller values. Few locations had
standing water thus large SWS during the wet period compared to few points with very small SWS
during the dry period owing to stronger demand by growing vegetation.
The generalized dimension, $D_q$ was subsequently used to characterize the scaling property and
variability in SWS (Fig. 6 and Fig. 7). The largest value of $f(q)$, referred to as the capacity
dimension ($D_0$) obtained at $q = 0$, was close to unity for all layers at different times (Fig. 6). The
information dimension ($D_1$) obtained at $q = 1$ was different from correlation dimension ($D_2$), the
average distribution density of the measurement for the surface layers in the wet period
(Grassberger and Procaccia, 1983). In this case, the different values of $D_0$, $D_1$ and $D_2$ indicated
multifractal nature of the distribution of SWS. Similarly, a non-unity value of $D_1/D_0$ (Montero,
2005) also indicated multifractal nature of SWS at the surface layer(s) during the wet period.
However, over the growing season, the $D_1$ and $D_2$ value approached closer to $D_0$ and indicated
monofractal type behavior. Similar values of $D_0$, $D_1$ and $D_2$ during the dry period also indicated
homogeneous distribution.
Joint multifractal distribution between the surface and various subsurface layers indicated the
similarity in the scaling patterns (Table 2). Basically, the hydrological processes of shallower
layers was more similar to the top layer, while deeper layers showed more observable disparities
from the surface. The nearest subsurface (20-40 cm) layer showed generally the highest similarity
with the surface (0-20 cm) layer.  However, in the wet period, the subsurface layers displayed the
smallest similarity to the surface layer, suggesting higher dynamic nature of hydrological
processes. In the dry period, stronger effect of vegetation overwhelmed the effect of small
variations, thus creating a more uniform distribution of SWS at all soil layers and showed stronger
similarity to the surface layers (Table 2).
Overall, our result revealed multifractal behavior of surface soil layers during the wet period
due to its dynamic nature. This behavior gradually changed with depth and time (Fig. 9). In the
deeper layers during the wet period, the behavior became less multifractal or nearly monofractal.
Similarly, in the dry period, the vegetation development and its high evapotranspirative demand



in semi-arid climate of the study area increasingly buffered the variation of SWS, as a result, all
the soil layers with less effect from environment forcing showed uniform distribution or
monofractal behavior (Fig. 9).
**5 Summary and Conclusions**
The transformation of information on soil water variability from one scale to another requires
knowledge on the scaling behaviour and the quantification of scaling index. Surface soil water can
be easily measured (e.g. remote sensing) and presents multi-scaling behaviour (requiring multiple
scaling indices). However, land-management practices requires the understanding of the
hydrological dynamics in the root zone and/or the whole soil profile. The scaling properties of the
surface soil layer can be used in the decision making provided the similar behavior holds at the
deep soil layer.
In this manuscript, the scaling properties of soil water storage at different soil layers measured
over five-year period were examined using multifractal and joint multifractal analysis. The scaling
properties of soil water storage mainly suggested monofractal scaling behavior. However, the
surface layer in the wet period or with high soil water storage tended to be multifractal in nature,
which gradually became monofractal with depth. With the decrease in soil water storage, the
scaling behavior became monofractal in nature at the later part of the year or growing season. The
year with high annual precipitation stored more water in the surface layer throughout the growing
period and displayed nearly multifractal scaling behavior. This multifractal nature indicated that
the transformation of information from one scale to another at the surface layer during the wet
period requires multiple scaling indices. On the contrary, the transformation requires single scaling
index during the dry period for the whole soil profile. The scaling properties of the surface layer
were highly correlated with that of the deep layers, which indicated a highly similar scaling
behaviour in the soil profile. The study was conducted in an undulating landscape from a semi-
arid climate and the results were very persistence over the years. Therefore, the observation
completed at the field scale in this type of landscape and climate may be generalized in similar
landscapes and climatic situations, otherwise may need to be examined thoroughly. The method
used here can be transferred to examine the scaling properties in other experimental situations.
**6 Acknowledgements**



The project was funded by the Natural Science and Engineering Research Council of Canada. The
help from the graduate student and the summer students of the Department of Soil Science at the
University of Saskatchewan in collecting field data is highly appreciated.

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

**Figure captions**
Fig. 1. Log-log plot between the aggregated variance of the SWS spatial series and the scale. A
linear relationship indicated the presence of scale invariance and scaling laws.
Fig. 2. Mass exponents for soil water storage spatial series measured at each 20 cm soil layer down
to 140 cm in 2008 and 2010 for a range of q (-15 to 15 at 0.5 increments). The solid line is a linear
reference created following the UM model of Schertzer and Lovejoy (1987) passing through (q =

0).

Fig. 3. Mass exponents for soil water storage spatial series from surface to different soil layers
(cumulative storage) at 20 cm increment down to 140 cm in 2008 and 2010 for a range of q (-15
to 15 at 0.5 increments). The solid line is a linear reference created following the UM model of
Schertzer and Lovejoy (1987) passing through (q = 0).
Fig. 4. Multifractal spectra of soil water storage spatial series measured at each 20 cm soil layer
down to 140 cm in 2008 and 2010 for a range of q (-15 to 15 at 0.5 increments).
Fig. 5. Multifractal spectra of soil water storage spatial series from surface to different soil layers
(cumulative storage) at 20 cm increment down to 140 cm in 2008 and 2010 for a range of q (-15
to 15 at 0.5 increments).
Fig. 6. Generalized dimension spectra of soil water storage spatial series measured at each 20 cm
soil layer down to 140 cm in 2008 and 2010 for a range of q (-15 to 15 at 0.5 increments).
Fig. 7. Generalized dimension spectra of soil water storage spatial series from surface to different
soil layers (cumulative storage) at 20 cm increment down to 140 cm in 2008 and 2010 for a range
of q (-15 to 15 at 0.5 increments).
Fig. 8: Conceptual schematics showing the vegetation growth patterns in the different section of
landscapes at different times of the year. The figure is developed based on field observations and
the scale is arbitrary.
Fig. 9: Conceptual schematics showing vegetation development over time, dominant water loss
processes and the scaling behavior of soil water storage at different depths. The figure is developed
based on field observations and scaling analysis. The scale of the figure is arbitrary.





**Tables**
Table 1. Maximum, minimum, and average soil water storage at different depths (20 cm increment) over the whole measurement period.

| | 0-20 cm | | | 20-40 cm | | | 40-60 cm | | | 60-80 cm | | | 80-100 cm | | | 100-120 cm | | | 120-140 cm | | |
|---|---|---|---|---|---|---|---|---|---|---|---|---|---|---|---|---|---|---|---|---|---|
| | Maximum (cm) | Minimum (cm) | Average (cm) | Maximum (cm) | Minimum (cm) | Average (cm) | Maximum (cm) | Minimum (cm) | Average (cm) | Maximum (cm) | Minimum (cm) | Average (cm) | Maximum (cm) | Minimum (cm) | Average (cm) | Maximum (cm) | Minimum (cm) | Average (cm) | Maximum (cm) | Minimum (cm) | Average (cm) |
| Jul 17 2007 | 13.96 | 3.25 | 5.65 | 11.55 | 3.09 | 5.63 | 9.43 | 2.59 | 5.73 | 9.06 | 3.34 | 5.90 | 9.51 | 3.22 | 5.89 | 9.81 | 3.55 | 6.05 | 9.81 | 3.54 | 6.14 |
| Aug 7 2007 | 13.96 | 3.05 | 4.90 | 9.28 | 2.73 | 5.04 | 8.30 | 2.40 | 5.21 | 9.36 | 2.75 | 5.48 | 8.23 | 2.96 | 5.57 | 7.52 | 3.17 | 5.62 | 9.11 | 3.17 | 5.67 |
| Sept 1 2007 | 13.96 | 2.26 | 5.29 | 9.28 | 3.00 | 5.08 | 8.08 | 2.42 | 5.23 | 6.98 | 2.75 | 5.38 | 7.17 | 2.92 | 5.52 | 8.08 | 3.20 | 5.64 | 9.07 | 3.23 | 5.73 |
| Oct 12 2007 | 8.30 | 3.40 | 5.04 | 6.92 | 3.07 | 5.03 | 6.74 | 2.43 | 5.19 | 7.60 | 2.81 | 5.36 | 8.39 | 2.93 | 5.48 | 7.92 | 3.25 | 5.60 | 8.55 | 3.25 | 5.67 |
| May 2 2008 | 13.96 | 4.49 | 6.28 | 9.96 | 4.09 | 6.03 | 9.43 | 3.69 | 5.80 | 8.83 | 3.16 | 5.74 | 9.51 | 2.90 | 5.66 | 9.81 | 3.26 | 5.70 | 9.81 | 3.30 | 5.75 |
| May 31 2008 | 13.96 | 3.30 | 5.21 | 9.28 | 1.54 | 5.51 | 8.08 | 1.58 | 5.55 | 6.85 | 3.00 | 5.58 | 7.08 | 3.08 | 5.64 | 8.08 | 3.22 | 5.70 | 8.39 | 3.25 | 5.79 |
| Jun 21 2008 | 8.77 | 3.06 | 4.70 | 7.84 | 3.43 | 5.25 | 6.86 | 2.80 | 5.38 | 6.78 | 2.77 | 5.52 | 7.08 | 3.04 | 5.61 | 7.73 | 3.28 | 5.69 | 8.48 | 3.23 | 5.77 |
| July 16 2008 | 7.07 | 2.78 | 4.03 | 6.78 | 3.06 | 4.77 | 6.71 | 2.60 | 5.10 | 6.75 | 2.56 | 5.30 | 6.84 | 2.91 | 5.43 | 6.98 | 3.17 | 5.56 | 7.01 | 3.16 | 5.64 |
| Aug 23 2008 | 4.96 | 2.44 | 3.40 | 5.66 | 2.73 | 4.11 | 6.02 | 2.37 | 4.59 | 6.44 | 2.36 | 4.90 | 6.56 | 2.63 | 5.12 | 6.85 | 3.04 | 5.30 | 6.81 | 2.99 | 5.42 |
| Sept 17 2008 | 4.64 | 2.66 | 3.51 | 5.63 | 2.79 | 4.07 | 5.91 | 2.49 | 4.55 | 6.28 | 2.45 | 4.85 | 6.59 | 2.63 | 5.05 | 6.68 | 3.05 | 5.25 | 6.91 | 2.96 | 5.37 |
| Oct 22 2008 | 6.11 | 3.83 | 4.96 | 6.03 | 3.10 | 4.37 | 5.92 | 2.52 | 4.53 | 6.13 | 2.46 | 4.79 | 6.55 | 2.63 | 5.00 | 6.61 | 3.00 | 5.18 | 6.73 | 1.22 | 5.28 |
| April 20 2009 | 13.96 | 4.73 | 6.67 | 11.55 | 3.62 | 5.84 | 10.49 | 3.23 | 5.62 | 8.83 | 2.97 | 5.48 | 9.51 | 2.67 | 5.38 | 9.81 | 3.08 | 5.49 | 9.81 | 2.85 | 5.66 |
| May 7 2009 | 13.96 | 4.45 | 5.97 | 9.51 | 3.68 | 5.70 | 8.08 | 3.26 | 5.49 | 8.30 | 3.00 | 5.36 | 7.85 | 2.73 | 5.35 | 9.81 | 3.01 | 5.43 | 8.91 | 2.84 | 5.51 |
| May 27 2009 | 12.60 | 3.67 | 5.43 | 8.15 | 3.55 | 5.52 | 8.08 | 3.43 | 5.39 | 6.78 | 3.13 | 5.37 | 7.16 | 2.64 | 5.39 | 8.08 | 2.96 | 5.51 | 8.45 | 2.80 | 5.53 |
| July 21 2009 | 6.92 | 3.16 | 4.56 | 7.24 | 3.16 | 4.83 | 6.55 | 2.91 | 5.00 | 6.72 | 2.95 | 5.23 | 6.77 | 2.58 | 5.24 | 6.91 | 3.02 | 5.34 | 6.89 | 3.24 | 5.43 |
| Aug 27 2009 | 6.64 | 3.42 | 5.01 | 6.67 | 3.57 | 5.07 | 6.32 | 2.84 | 4.92 | 6.50 | 2.85 | 5.03 | 6.76 | 2.57 | 5.16 | 6.79 | 3.00 | 5.25 | 6.90 | 3.02 | 5.34 |
| Oct 27 2009 | 6.65 | 3.89 | 5.30 | 6.44 | 3.44 | 4.90 | 6.04 | 2.74 | 4.80 | 6.36 | 2.68 | 4.91 | 6.55 | 2.60 | 5.05 | 6.71 | 3.05 | 5.17 | 6.71 | 2.79 | 5.29 |
| April 6 2010 | 13.96 | 4.67 | 6.47 | 9.51 | 3.53 | 5.52 | 9.43 | 3.19 | 5.31 | 8.83 | 2.91 | 5.35 | 9.51 | 2.61 | 5.23 | 9.81 | 3.01 | 5.34 | 9.81 | 2.83 | 5.41 |
| May 19 2010 | 13.96 | 4.08 | 6.04 | 11.32 | 4.28 | 5.94 | 10.49 | 4.46 | 5.94 | 8.75 | 4.08 | 5.93 | 8.60 | 3.55 | 5.90 | 9.81 | 4.03 | 5.91 | 9.81 | 3.96 | 5.85 |
| June 14 2010 | 13.96 | 4.38 | 6.54 | 11.55 | 4.48 | 6.32 | 10.49 | 4.58 | 6.31 | 8.83 | 4.27 | 6.29 | 9.51 | 3.86 | 6.22 | 9.81 | 4.37 | 6.24 | 9.81 | 4.50 | 6.20 |
| Sept 28, 2010 | 13.96 | 4.51 | 6.33 | 11.55 | 4.48 | 6.16 | 9.43 | 3.77 | 6.08 | 8.83 | 3.91 | 6.13 | 9.51 | 3.83 | 6.12 | 9.81 | 4.11 | 6.16 | 9.79 | 4.18 | 6.20 |
| May 13, 2011 | 13.96 | 4.82 | 7.12 | 11.55 | 4.87 | 6.61 | 10.49 | 4.75 | 6.50 | 9.21 | 4.54 | 6.40 | 9.51 | 4.16 | 6.34 | 9.96 | 3.17 | 6.32 | 9.79 | 4.30 | 6.45 |
| Jun 6, 2011 | 13.96 | 4.31 | 7.05 | 11.55 | 4.56 | 6.59 | 10.49 | 3.85 | 6.52 | 9.06 | 4.75 | 6.44 | 9.51 | 4.21 | 6.40 | 9.96 | 3.17 | 6.39 | 9.79 | 4.77 | 6.52 |
| Jun 29, 2011 | 13.96 | 4.93 | 7.16 | 11.55 | 4.96 | 6.73 | 10.49 | 4.29 | 6.64 | 9.74 | 4.42 | 6.57 | 9.51 | 4.28 | 6.49 | 9.96 | 3.17 | 6.46 | 9.79 | 4.30 | 6.55 |
| Sept 29, 2011 | 12.60 | 3.11 | 5.25 | 8.15 | 3.46 | 5.50 | 8.08 | 2.88 | 5.68 | 7.58 | 4.03 | 5.82 | 9.19 | 3.77 | 5.89 | 9.51 | 3.81 | 6.02 | 9.36 | 4.14 | 6.04 |






Table 2: Correlation between joint multifractal coefficients of surface to different subsurface
layers measured at 20 cm interval in 2008.

|  | 2 May 2008 | 31 May 2008 | 21 Jun. 2008 | 16 Jul. 2008 | 23 Aug. 2008 | 17 Sep. 2008 | 22 Oct. 2008 |
|---|---|---|---|---|---|---|---|
| 0-20 cm vs. 20-40 cm | 0.96 | 0.98 | 0.99 | 0.99 | 0.99 | 1.00 | 1.00 |
| 0-20 cm vs. 40-60 cm | 0.93 | 0.96 | 0.96 | 0.97 | 0.97 | 1.00 | 1.00 |
| 0-20 cm vs. 60-80 cm | 0.93 | 0.94 | 0.95 | 0.95 | 0.96 | 0.99 | 0.99 |
| 0-20 cm vs. 80-100 cm | 0.92 | 0.92 | 0.93 | 0.94 | 0.94 | 0.98 | 0.99 |
| 0-20 cm vs. 100-120 cm | 0.92 | 0.92 | 0.93 | 0.93 | 0.93 | 0.97 | 0.99 |
| 0-20 cm vs. 120-140 cm | 0.93 | 0.94 | 0.95 | 0.94 | 0.94 | 1.00 | 1.00 |

















**Figures**



Figure 1





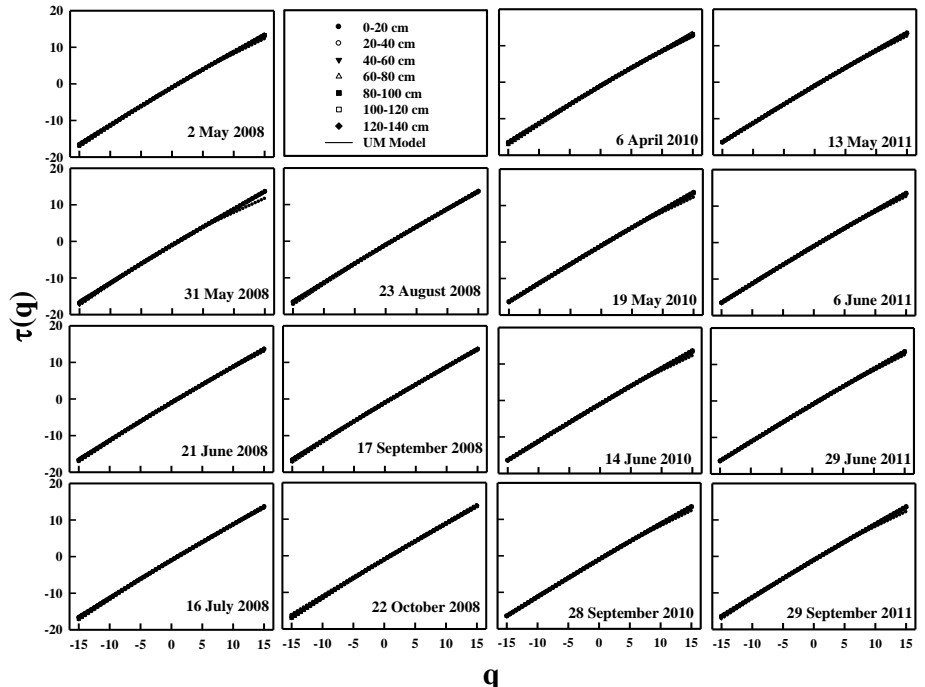

Figure 2

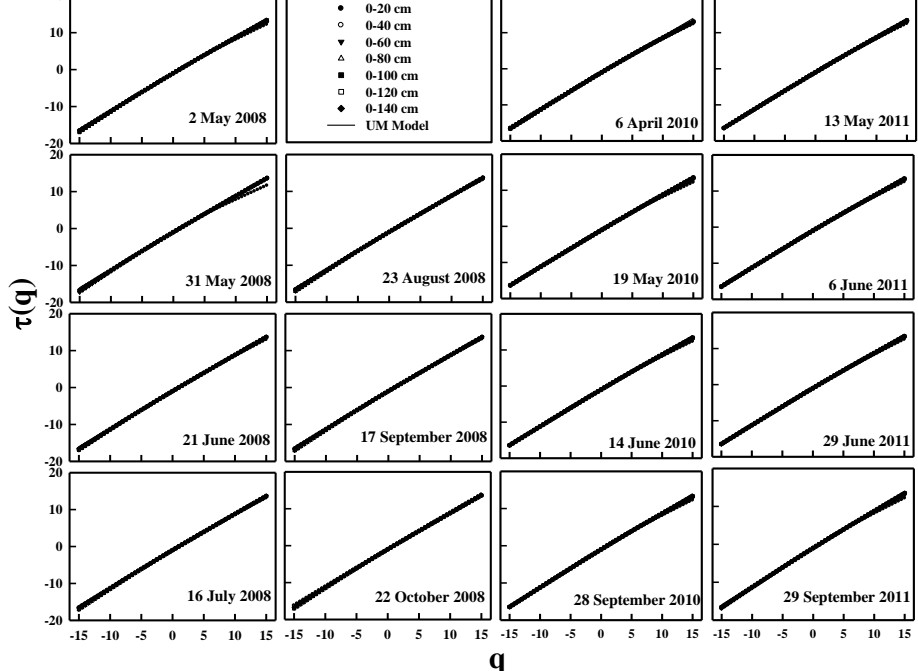

Figure 3





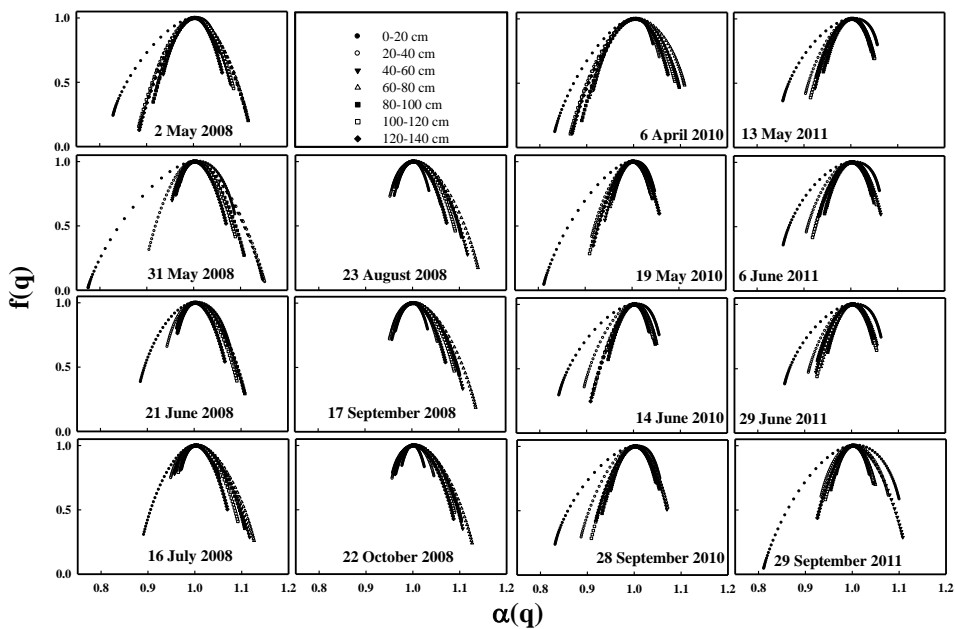

Figure 4

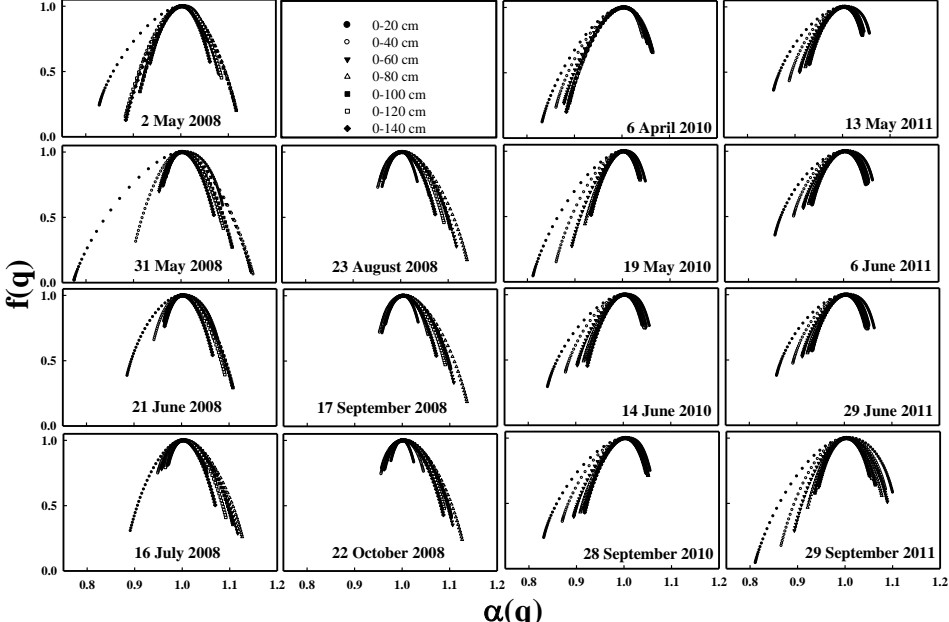

Figure 5



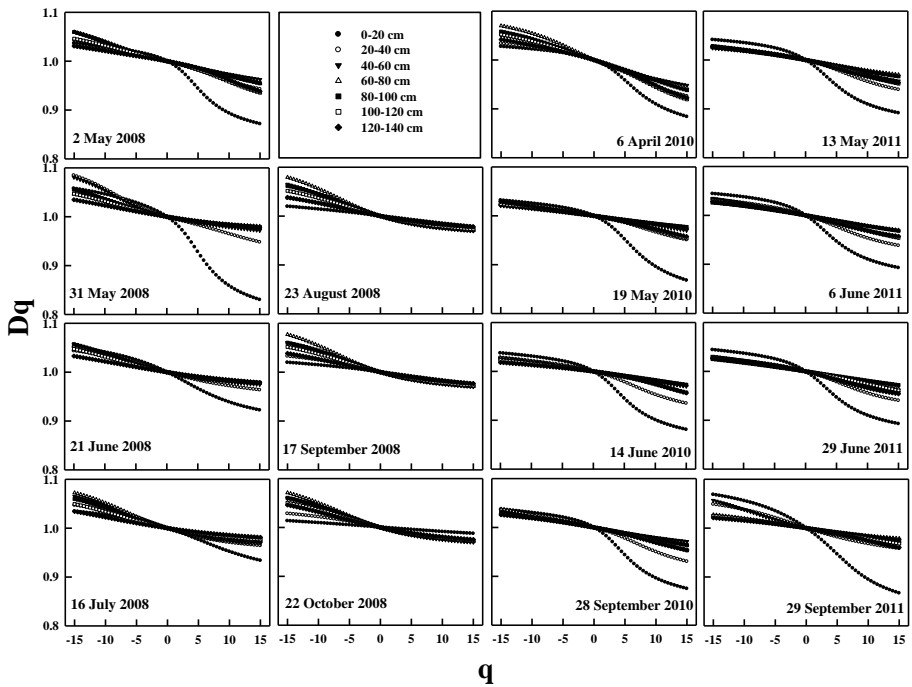

Figure 6

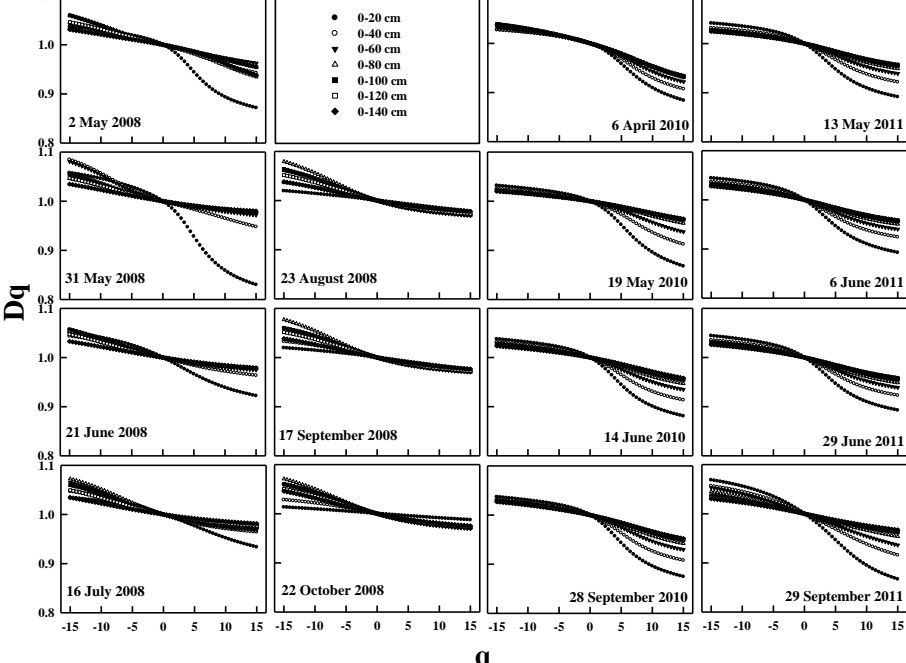

Figure 7





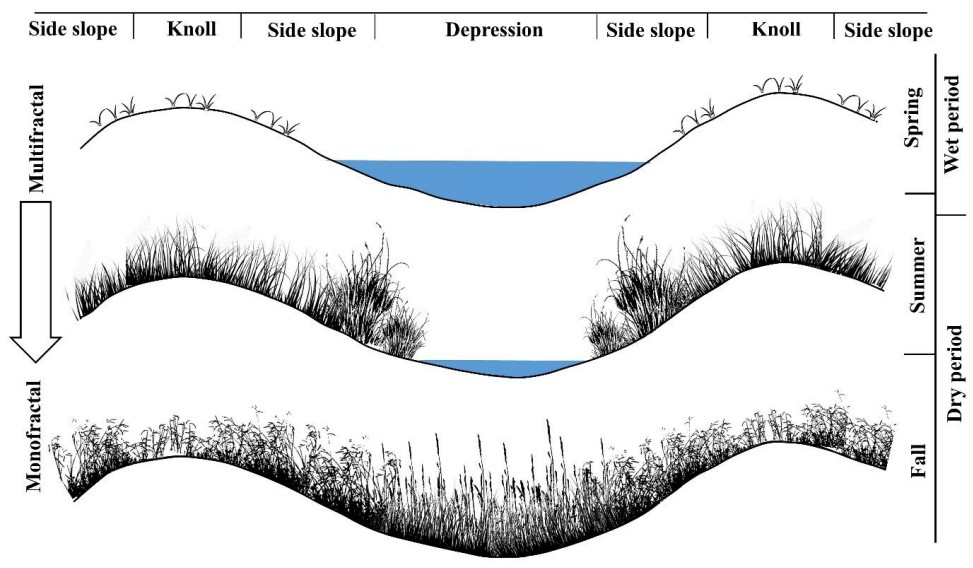

Figure 8

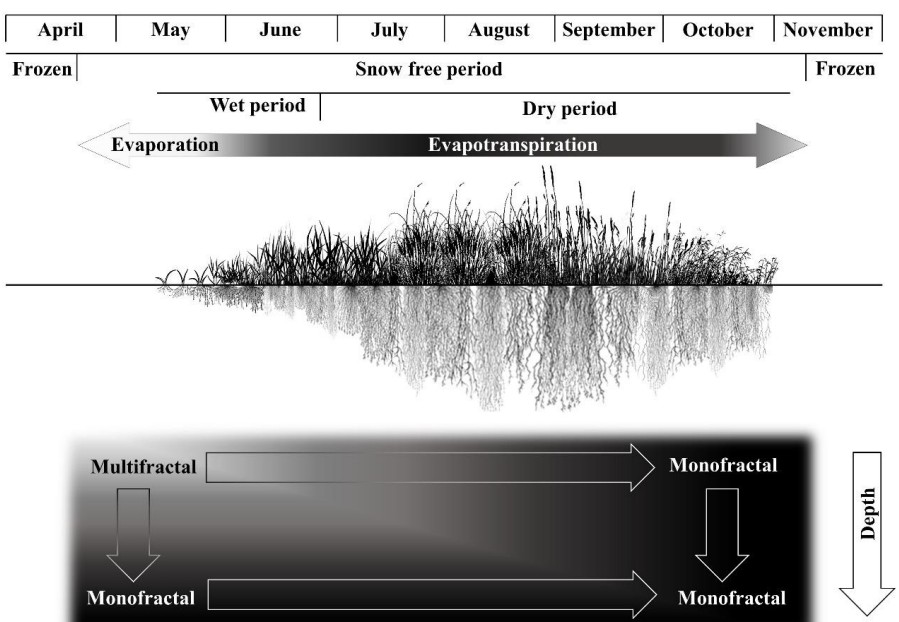

Figure 9