# Peer review of "Fractal behavior of soil water storage at multiple depths"

_Nonlinear Processes in Geophysics, 2015_

## Referee Comment (RC1) · Anonymous Referee #1 · 7 Mar 2016

General Comments

This manuscript repeatedly evaluates soil water storage (SWS) across an transect consisting of 128 individual measurement points by multifractal and joint multifractal analysis. In each point SWS was determined at seven depth increments, and this several times per year during a five years period. Therefore the relationship between several multifractal parameters obtained from SWS transects as a function of depth and time were investigated. This study follows previous studies carried out with in the same site, with the same experimental design and using similar methods of analysis. The rationale and the objectives exposed in the Introduction are worthwhile and in general the work appears well justified and appealing for the international reader of this journal. The main findings, such as the usefulness of either multifractal or monofractal parameters to assess patters of heterogeneity and evenness of SWS transects with increasing

soil depth and for different seasons of the year. In general, the paper is well written and organized, and represents an original contribution, even if it follows previous similar work. The results are based in robust data analysis. This study also is compatible with the aims Nonlinear Processes in Geophysics and may fit well into the scope of the current special section titled "Multifractal analysis in soil systems". In my opinion, it should be acceptable for publication following minor to moderate revisions.

In my opinion the manuscript could be ameliorated by plotting selected multifractal parameters (for example, the amplitude of the singularity spectrum, ïĄąmax-ïĄąmin, or the information dimension D1) as a function of depth and time, or both, depth and time. Next I'm indicating two places where plots are recommended, but this is not exhaustive.

Specific Comments 1. Page 4, Lines 95-100. I recommend to briefly describe the methods used to measure soil water content and to evaluate soil water storage (SWS), even if they have been already detailed described before. 2.- Page 9, Lines 226-243. I suggest to draw a graphic with these statistical information; then check if including this graph increases readability. 3.- Page 12 and 13, Lines 342-352. Again, I suggest to draw a graphic plotting (ïĄąmax-ïĄąmin) as a function of depth for several mesarument periods. 4.- Page 23, Table 2. I suggest to include figures showing some multifractal spectra either in the main manuscript or as supplementary content. 5.- Page 25, Figures 2 and 3. I recommend to show only two or four selected plots of mass exponent functions to increase visibility. (because of thee small size of the Figures, differences are hardly to vew). 6.- Page 26, Figures 4 and 5. I suggest to take into account the shape of the singularity spectra and not only the amplitude in ths and the Results and in the Discussion sections; also these shapes should provide valuable information, I guess. 7.- Page 28, Figure 8. I suggest to move Figure 8 (scheme of the vegetation growth patterns) either to the Material and Method section, or to the section 3.1 (Spatial pattern of soil water storage at different depths). Indeed this Figure is related to the Discussion section, but it is also pertinent to previous section.

---

## Editor Comment (EC1) · J. M. Miras Avalos (Editor) · 14 Mar 2016

The manuscript entitled "Fractal behavior of soil water storage at multiple depths" (Reference number NPG-2015-81) authored by W. Ji, M. Lin, A. Biswas, B.C. Si, H.W. Chau, and P. Cresswell presents results from a five-year study on the soil water storage from a transect in a hummocky landscape of central Canada. The authors applied multifractal and joint multifractal theories to this huge dataset in order to describe the fractal behavior of this variable at different depths along the transect.

I agree with the comments posted by reviewer1 and consider this manuscript very-well written and acceptable for publication after several modifications. Unfortunately, I have not received the comments from the second reviewer yet. Anyway, I carefully read the submitted manuscript and performed some comments and suggestions in order to

improve its quality.

The reported work is interesting and fits perfectly well within the scope of the Special Issue "Multifractal analysis in soil systems" to be published in Nonlinear Processes in Geophysics. However, the manuscript is rather long and information can be condensed as well as reduced since it seems repetitive in some portions. Tables can be improved and, from my point of view, figure 9 is not needed and can be deleted. Finally, a few English mistakes must be corrected.

In the attached file (supplement), I provide the authors with some suggestions in order to improve their manuscript. Therefore, the authors must address these issues prior to the acceptance of their manuscript. They must correct them in order that this manuscript achieves the standard quality for being published in Nonlinear Processes in Geophysics.

Therefore, I recommend a moderate revision prior to its publication in this journal.

Please also note the supplement to this comment:
http://www.nonlin-processes-geophys-discuss.net/npg-2015-81/npg-2015-81-EC1-supplement.pdf

**Supplement:**

**The manuscript entitled "Fractal behavior of soil water storage at multiple depths" (Reference number NPG-2015-81) authored by W. Ji, M. Lin, A. Biswas, B.C. Si, H.W. Chau, and P. Cresswell presents results from a five-year study on the soil water storage from a transect in a hummocky landscape of central Canada. The authors applied multifractal and joint multifractal theories to this huge dataset in order to describe the fractal behavior of this variable at different depths along the transect.**

**The reported work is interesting and fits perfectly well within the scope of the Special Issue "Multifractal analysis in soil systems" to be published in Nonlinear Processes in Geophysics. However, the manuscript is rather long and information can be condensed as well as reduced since it seems repetitive in some portions. Tables can be improved and, from my point of view, figure 9 is not needed and can be deleted. Finally, a few English mistakes must be corrected.**

**In the following lines, I provide the authors with some suggestions in order to improve their manuscript. Therefore, the authors must address these issues prior to the acceptance of their manuscript. They must correct them in order that this manuscript achieves the standard quality for being published in *Nonlinear Processes in Geophysics*.**

**Therefore, I recommend a moderate revision prior to its publication in this journal.**

**Specific comments to the authors:**

*Abstract:*

*Line 16: "hold at deep layer. Current study" could be changed to "are kept at deeper layers. The current study".*

*Line 17: "its relationship", shouldn't it be "their relationships"?*

*Line 18: Remove the word "the" before "supporting" and "plant root".*

*Line 20: Please, indicate here the years and the study site.*

*Line 21: I would remove "with large SWS". I would use "for multiple scaling" instead of "of multiple scaling".*

*Line 23: "for a single scaling" instead of "of single scaling".*

*Lines 24-25: "The dynamic nature..." Please, re-phrase this sentence, since it is confusing.*

*Line 26: "of the growing season", of what? I would also remove "with low SWS".*

*Lines 28-30: Only for the dry period? This is somewhat unclear.*

*Keywords:*

*I would use only one, either "scaling" or "scale invariance".*

*Introduction:*

*This section is well-written and provides enough information about the background of the presented work; however, it looks rather repetitive. Could you condense some information, please?*

*Lines 44-46: You used the word "scale" too many times in this sentence.*

*Line 53: "a typical of scaling process", a typical what?*

*Line 55: "using the fractal theory" instead of "using fractal theory".*

*Line 57: Remove the word "scaling" between "single" and "coefficient".*

*Line 58: Remove the word "scaling" between "monofractal" and "behaviour"; by the way, should it be spelled like that or "behavior" as you used in the title?*

*Line 60: "(multifractals) for quantifying its spatial" instead of "(multifractal scaling) in quantifying spatial".*

*Lines 62-63: Please, remove this sentence because it is not needed since this statement is implied in the next sentences.*

*Line 65: "and drying cycles have been reported" instead of "and drying has been reported".*

*Line 67: "can provide a quick estimate" instead of "can provide an estimate".*

*Line 68: Remove the word "quickly".*

*Lines 68-69: I am not sure that "indicating the superficial properties" is needed.*

*Line 71: "is the most dynamic in nature" instead of "are most dynamic in nature".*

*Line 72: Please, check English in "the observed scaling properties holds for the deep layers".*

*Line 78: "to examine over time the scaling properties" instead of "to examine the scaling properties".*

*Line 79: Remove "over time".*

*Lines 80-81: "multiple depth layers and at soil layers with increasing depth from the surface (cumulative depth)", this is somewhat unclear. Please, re-phrase.*

*Line 83: Remove the word "layer" after "surface".*

*Materials and Methods:*

*Line 87: Please, indicate the elevation above sea level of the study site.*

*Line 89: "differently" instead of "different".*

*Line 93: "extending in the north-south" instead of "extending in north-south".*

*Line 95: "intervals" instead of "interval".*

*Line 96: "at every 20 cm depth", you should indicate down to what depth.*

*Line 97: Remove "of" after "depths".*

*Line 112: "and better characterize its spatial variability" instead of "and characterize its spatial variability better".*

*Lines 113-114: I think that this is not needed and can be removed.*

*Line 119: Remove "(scaling)".*

*Line 131: "with their associated masses" instead of "with its associated mass".*

*Line 135: "to characterize", to characterize what?*

*Lines 135-138: This last sentence is not clear, please, re-phrase it.*

*Line 146: Define UM when first used, please.*

*Line 147: "a reference line that represented" instead of "a linear reference line which represented".*

*Line 153: "intervals" instead of "interval".*

*Line 155: "was also completed" instead of "were also completed".*

*Line 156: "was checked" instead of "was tested".*

*Line 158: "proven" instead of "proved".*

*Lines 174-175: From my point of view you could delete this sentence "This spectrum also enabled us to examine the local scaling property".*

*Lines 181-184: Here, you talk about models but I think you should use indicators or indices instead.*

*Line 197: "When this value equals to 1" instead of "The value equal to 1".*

*Line 202: "in size $\varepsilon$" should be "of size $\varepsilon$".*

*Line 210: "is the partition function" instead of "is partition function".*

*Line 219: Remove "value of".*

*Lines 220-221: "Pair value...", this sentence is unclear. Please, re-phrase it.*

*Results:*

*Lines 226-229: These values are not included in Table 1. Why did you mention this table here?*

*Lines 230-231: "The highest average SWS..." this is not true for all depths.*

*Line 231: "large amount of spring rainfall", data on rainfall are not shown.*

*Line 233: "summers" instead of "summers". Besides, this is not true for all depths.*

*Line 236: "wider" instead of "bigger".*

*Lines 235-240: Please, re-phrase, this is rather confusing.*

*Line 241: "coefficients of variation" instead of "coefficient of variations".*

*Line 245: The minimum is 6.71 cm according to table 1 and not 6.72 cm as you said in the text.*

*Line 250: "field-average because they were situated" instead of "field-average and were situated".*

*Line 257: Remove the word "even".*

*Line 260: "decreased" instead of "decrease".*

*Lines 267-268: I would remove the last sentence because this statement is logical since coefficient of variation and standard deviation are related variables.*

*Lines 270-271: This first sentence is not needed and can be removed.*

*Lines 271-273: This looks like materials and methods and not results.*

*Line 275: Remove "The scale invariance" and substitute it for "which".*

*Line 277: I would use "dates" instead of "measurements".*

*Line 279: What do you mean by "soil layers with cumulative depths".*

*Lines 281-282: Define UM and SSR when first used, please.*

*Line 291: Remove "the" before "deep layers".*

*Line 298: "increasing" instead of "the increase of".*

*Line 302: "went" instead of "going".*

*Line 304: Remove the word "statically".*

*Line 305: "of the first three" instead of "of first three".*

*Lines 321-329: This is messy and unclear, even somewhat repetitive. Please, re-phrase.*

*Lines 330-331: I would re-phrase this sentence to "The height of the multifractal spectrum at*

*different depths was very similar over time".*

*Line 333: "smaller with depth" instead of "smaller at depths".*

*Line 339: "on 21 June 2008" instead of "measurement of 21 June 2008".*

*Line 343: "in the first three" instead of "in first three".*

*Line 356: Remove the word "all".*

*Line 371: "A very similar trend was observed in other years". These data are not shown. Indicate this and also briefly specify the similarity.*

*Discussion:*

*This section can be reduced since it seems repetitive and information can be condensed because some paragraphs look like materials and methods.*

*Line 373: "in the soil" instead of "in soil layers".*

*Line 377: "within a short period" instead of "within short period".*

*Line 378: "contributes" instead of "contribute".*

*Line 384: Remove one dot at the end of the sentence.*

*Line 389: "of water" instead of "of the water".*

*Line 390: "stored a larger" instead of "stored larger".*

*Line 394: Remove one dot at the end of the sentence.*

*Line 396: "average SWS in a year", only in one year?*

*Line 397: "compared" instead of "comparing".*

*Line 400-402: "Stronger demand…" This sentence is not clear, please, re-phrase it.*

*Line 404: "deeper" instead "deep".*

*Line 405: I would use "factors" or "forces" instead of "forcing".*

*Line 406: I would remove "and was very dynamic in nature".*

*Line 412: "the presence of scaling laws" can be changed to "that SWS behaved under scaling laws".*

*Lines 413-418: This portion looks like materials and methods.*

*Line 430: "factors" instead of "forcing". "and this exhibited a monofractal" instead of "and*

*exhibited monofractal".*

*Line 435: I would remove "to describe scaling property".*

*Lines 439-450: This looks like materials and methods. In addition, it is quite repetitive.*

*Lines 451-457: Is this paragraph really needed? It repeats the former paragraphs.*

*Line 466: "Surface one or two layers", do you mean the two upper soil layers?*

*Lines 469-471: This last sentence is rather confusing. Please, re-phrase it.*

*Lines 476-477: I do not understand what you mean here.*

*Line 479: "indicated the multifractal" instead of "indicated multifractal".*

*Line 480: Remove the word "closer".*

*Line 481: "indicated a monofractal" instead of "indicated monofractal".*

*Line 482: "distributions" instead of "distribution".*

*Line 485: "layers were similar to those of the top layer" instead of "layers was more similar to the top layer". Besides, what do you mean by "showed more observable"?*

*Line 488: "a higher dynamic" instead of "higher dynamic".*

*Line 489: "a stronger effect" instead of "stronger effect".*

*Line 490: "small variations" of what? Besides, "soil layers that showed a stronger" instead of "soil layers and showed stronger".*

*Line 492: "our results revealed a multifractal" instead of "our result revealed multifractal".*

*Line 493: "due to its dynamic nature" of what? The wet period? The soil? The SWS? Not clear.*

*Line 496: "in the semi-arid" instead of "in semi-arid".*

*Line 497: "environmental factors showed a uniform" instead of "environment forcing showed uniform".*

*Line 498: Is figure 9 really needed?*

*Conclusions:*

*Line 501: "scaling indices" instead of "scaling index".*

*Line 503: "require" instead of "requires".*

*Lines 504-506: The idea mentioned in this last sentence is not well developed throughout the text,*

*especially it is not discussed at all in the discussion section. However, it appeared in the abstract.*

*Line 508: "over a five-year" instead of "over five-year".*

*Line 509: "suggested a monofractal" instead of "suggested monofractal".*

*Line 512: Remove "in nature".*

*Lines 513-514: Please, check English on this sentence.*

*Line 516: "requires a single scaling" instead of "requires single scaling".*

*Line 520: "very persistence"? Do you mean "consistent"?*

*Lines 520-522: "Therefore, the observation completed…", I am not sure about this conclusion.*

*References:*

*Lines 544-545: Use the abbreviated title for the journal "Phys. Rev. Lett."*
*Line 556: Use the abbreviated title for the journal "Phys. Rev. Lett."*
*Lines 558-559: Why is the title of the article in capital letters?*
*Line 565: Use the abbreviated title for the journal "Remote Sens. Environ."*
*Lines 569-570: Use the abbreviated title for the journal "Remote Sens. Environ."*
*Line 584: Use the abbreviated title for the journal "Phys. Rev."*
*Lines 596-597: Why is the title of the article in capital letters?*
*Line 600: Use the abbreviated title for the journal "J. Geophys. Res."*

*Figure captions*
*In the caption for the first figure include "for three selected dates".*
*Lines 619-636: In these figures data from 2011 are also shown; however, you indicate 2008 and 2010 in the caption. Please, include also 2011.*
*Lines 640-642: Is figure 9 really needed?*

*Table 1: Please, include the five-year averages, since you refer to them in the text.*
*Table 2: Please, indicate the number of data used for each correlation. Was it the same for all dates and depths? I would re-phrase the title of this table to "Correlation coefficients between joint multifractal indices (α and β) of the surface layer with those from subsurface layers at 20 cm*

*intervals in 2008".*

*Figures:*

*Figure 1: Why not showing the Y-scale in all left graphs?*

*Figure 2 and 3: It is very difficult to distinguish the points from each depth. Besides, the UM model is missing from the graphs.*

*Figures 6 and 7: Some values are overlapped in the Y-axis.*

*Figure 9: Is this figure really needed?*

---

## Referee Comment (RC2) · Anonymous Referee #2 · 16 Mar 2016

This work makes use of multifractal analysis and joint multifractal analysis to study the spatio-temporal behavior of soil water storage (SWW) at multiple depths. Several interested implications about the scaling nature of SWW that are relevant to transfer information from one scale to another, are shown.

The manuscript is well structured and conclusions are drawn from sound mathematical theories applied to a rich enough database consistent with the algorithms used to estimate theoretical parameters. Therefore, I recommend acceptance following minor revisions. My specific comments are itemized below:

page 6, line 163 and page 6, line 167: I suggest to use Chhabra and Jensen (1989) as a reference instead of Everest and Mandelbrot (1992).

page 7, line 197 and page 17, line 478: This parameter was first introduced in Caniego,

Martín and San José (2003)

page 14, line 384: There are two points instead of one to the end of the line.

Reference: Caniego, F.J., Martín, M.A. and San José, F., 2003. Rényi dimensions of soil pore size distribution. Geoderma 112 (2003) 205– 216.

---

## Editor Comment (EC2) · J. M. Miras Avalos (Editor) · 17 Mar 2016

Dear authors,

I am glad to inform you that two referees have reviewed your manuscript entitled "Fractal behavior of soil water storage at multiple depths" (Reference number NPG-2015-81). Both referees agree that the reported work is interesting and fits perfectly well within the scope of the Special Issue "Multifractal analysis in soil systems" to be published in Nonlinear Processes in Geophysics. However, they pointed out minor to moderate remarks on your manuscript that prevent its publication in its present form. I agree with the comments of both referees and I also reviewed your manuscript (please, see the attached file on my previous message). Therefore, you must address the issues raised by the reviewers prior to the acceptance of your manuscript in this Special Issue

of Nonlinear Processes in Geophysics. Finally, I recommend a moderate revision prior to publication in this Special Issue.

Sincerely, José Manuel Mirás-Avalos
* * *

---

## Author Comment (AC3) · 26 May 2016

Please see the response to comments in the next comment thread.

---

## Author Response (AR1)

**Comments from Referees: J. M. Miras Avalos (Editor)**

The manuscript entitled "Fractal behavior of soil water storage at multiple depths" (Reference number NPG-2015-81) authored by W. Ji, M. Lin, A. Biswas, B.C. Si, H.W. Chau, and P. Cresswell presents results from a five-year study on the soil water storage from a transect in a hummocky landscape of central Canada. The authors applied multifractal and joint multifractal theories to this huge dataset in order to describe the fractal behavior of this variable at different depths along the transect.

I agree with the comments posted by reviewer1 and consider this manuscript very-well written and acceptable for publication after several modifications. Unfortunately, I have not received the comments from the second reviewer yet. Anyway, I carefully read the submitted manuscript and performed some comments and suggestions in order to improve its quality.

The reported work is interesting and fits perfectly well within the scope of the Special Issue "Multifractal analysis in soil systems" to be published in Nonlinear Processes in Geophysics. However, the manuscript is rather long and information can be condensed as well as reduced since it seems repetitive in some portions. Tables can be improved and, from my point of view, figure 9 is not needed and can be deleted. Finally, a few English mistakes must be corrected.

In the attached file (supplement), I provide the authors with some suggestions in order to improve their manuscript. Therefore, the authors must address these issues prior to the acceptance of their manuscript. They must correct them in order that this manuscript achieves the standard quality for being published in Nonlinear Processes in Geophysics.

Therefore, I recommend a moderate revision prior to its publication in this journal.

Please also note the supplement to this comment: http://www.nonlin-processes-geophys-discuss.net/npg-2015-81/npg-2015-81-EC1- supplement.pdf

*-- Response: Thank you very much for your detailed comments. Your comments and suggestions have greatly help improve our manuscript. We have tried to condense the context by deleting some repetitive parts and combining the information by figures. The tables and figures have been revised and supplemented according to all the review comments. New figures are also added and figure sequence have been changed to improve the structure of manuscript. We have also worked carefully on the English.*

*We have also documented all the changes we made in the revised version and responded to each comment individually.*

**Author's changes**

Lines 28-30: Only for the dry period? This is somewhat unclear.

*-- Response: Not only dry period, here it is just to discuss the dry period specifically as a comparison to wet period mentioned above. We changed the "in contrast" to "on the other hand", which may shows the logic better. (L26)*

Lines 44-46: You used the word "scale" too many times in this sentence.

*-- Response: "Scale" has been changed to "extent" in some sentences to increase the variability according your suggestions. (L40, L47, etc)*

Lines 68-69: I am not sure that "indicating the superficial properties" is needed.

*-- Response: We think it is required, for it acknowledges the achievement of previous studies. (L71)*

**Materials and Methods:**

Line 87: Please, indicate the elevation above sea level of the study site.

*-- Response: The elevation data included. (L89)*

Line 96: "at every 20 cm depth", you should indicate down to what depth.

*-- Response: Has added "down to 140cm" (L104)*

Lines 135-138: This last sentence is not clear, please, re-phrase it

*-- Response: The sentence has been rewritten (L150-152)*

Lines 226-229: These values are not included in Table 1. Why did you mention this table here?

*-- Response: Yes, that was a mistake. We have deleted this.*

Lines 230-231: "The highest average SWS…" this is not true for all depths.

*-- Response: The clause of "for the surface layer" has been added. (L257)*

Line 231: "large amount of spring rainfall", data on rainfall are not shown.

*-- Response: Data is added. (L258)*

Line 245: The minimum is 6.71 cm according to table 1 and not 6.72 cm as you said in the text

*-- Response: The text has been changed to 6.71cm as it was a typo. (l273)*

Line 146: Define UM when first used, please.

*-- Response: The UM is expanded during first use. (L160)*

Lines 321-329: This is messy and unclear, even somewhat repetitive. Please, re-phrase.

*-- Response: We have modified the paragraph and tried to make it clearer. (L347-359)*

Line 371: "A very similar trend was observed in other years". These data are not shown. Indicate this and also briefly specify the similarity.

*-- Response: We have added a new figure (Fig. 11) to explain the pattern.*

Line 396: "average SWS in a year", only in one year?

*-- Response: No, it was for other years too. This is a general observation. We have deleted the word 'in a year'. (L445)*

Lines 271-273: This looks like materials and methods and not results.

*-- Response: It reads like that as the result was generalized. This is necessary to main the flow of the results and discussion. So, we kept this section. However, we did modify a bit. (L484-496)*

Discussion

This section can be reduced since it seems repetitive and information can be condensed because some paragraphs look like materials and methods.

*-- Response: We have reorganized the paragraphs and reinterpreted the figures to provide further information after the analysis in the result section.*

Lines 451-457: Is this paragraph really needed? It repeats the former paragraphs.

*-- Response: We think this summary is required to increase the readability. Therefore, we kept this paragraph.*

Line 498: Is figure 9 really needed?

*-- Response: We think the figure is a summary of the main conclusion which is needed for the ease of understanding. This also shows the general trend in the data and the analysis result. Therefore, we kept the figure in the revised manuscript.*

Lines 504-506: The idea mentioned in this last sentence is not well developed throughout the text, 7 especially it is not discussed at all in the discussion section. However, it appeared in the abstract.

*-- Response: Yes, it seemed little weak and we have deleted this sentence.*

Lines 520-522: "Therefore, the observation completed…", I am not sure about this conclusion.

*-- Response: Through this sentence, we mean to say the general pattern of the scaling indices. The relationship can be directly transferred to other field situations given the similar kind of landscape and climate condition.*

Table 1: Please, include the five-year averages, since you refer to them in the text.

*-- Response: Added.*

Table 2: Please, indicate the number of data used for each correlation. Was it the same for all dates and depths?

*-- Response: The number of data is same for all and is mentioned in the title.*

Figure 1: Why not showing the Y-scale in all left graphs?

*-- Response: We have added y-axes for all the plots.*

Figure 2 and 3: It is very difficult to distinguish the points from each depth. Besides, the UM model is missing from the graphs.

*-- Response: Yes, there were too many graphs. We have reduced to only three. All the graphs have the UM model but it is not visible due to the condition of the plots.*

Figures 6 and 7: Some values are overlapped in the Y-axis.

*-- Response: We have modified the Fig. 6 and Fig 7. These are assigned to new numbers.*

Figure 9: Is this figure really needed?

*-- Response: We think the figure is a summary of the main conclusion which is needed for the ease of understanding. This also shows the general trend in the data and the analysis result. Therefore, we kept the figure in the revised manuscript.*

**Comments from Referees: Anonymous Referee #1**

**General Comments**: This manuscript repeatedly evaluates soil water storage (SWS) across an transect consisting of 128 individual measurement points by multifractal and joint multifractal analysis. In each point SWS was determined at seven depth increments, and this several times per year during a five years period. Therefore the relationship between several multifractal parameters obtained from SWS transects as a function of depth and time were investigated. This study follows previous studies carried out with in the same site, with the same experimental design and using similar methods of analysis. The rationale and the objectives exposed in the Introduction are worthwhile and in general the work appears well justified and appealing for the international reader of this journal. The main findings, such as the usefulness of either multifractal or monofractal parameters to assess patters of heterogeneity and evenness of SWS transects with increasing soil depth and for different seasons of the year. In general, the paper is well written and organized, and represents an original contribution, even if it follows previous similar work. The results are based in robust data analysis. This study also is compatible with the aims Nonlinear Processes in Geophysics and may fit well into the scope of the current special section titled "Multifractal analysis in soil systems". In my opinion, it should be acceptable for publication following minor to moderate revisions. In my opinion the manuscript could be ameliorated by plotting selected multifractal parameters (for example, the amplitude of the singularity spectrum, ï ¿ A ¿ amax-ï ¿ A ¿ amin, or the information dimension D1) as a function of depth and time, or both, depth and time. Next I'm indicating two places where plots are recommended, but this is not exhaustive.

*Response: Thank you very much for your detailed comments. It really helped modify the manuscript. We have addressed the comments individually and documented the responses below.*

**Specific Comments** 1. Page 4, Lines 95-100. I recommend to briefly describe the methods used to measure soil water content and to evaluate soil water storage (SWS), even if they have been already detailed described before.

*Response: We have included a brief description of the data collection in the materials an methods (L102-111).*

2.- Page 9, Lines 226-243. I suggest to draw a graphic with these statistical information; then check if including this graph increases readability.

*Response: We have included a new figure (Fig. 11) showing the joint multifractal spectra between two spatial series of soil water storage measure on 22 October 2008.*

3.- Page 12 and 13, Lines 342-352. Again, I suggest to draw a graphic plotting (ï ¿ A ¿ amax-ï ¿ A ¿ amin) as a function of depth for several mesarument periods.

*Response: Thanks for the suggestion. We have included two figures; Fig. 5 showing the $\alpha_{max}$-$\alpha_{min}$ values for all the measurements at all depths and Fig. 8 showing the D1 values for all the measurements at all depths.*

4.- Page 23, Table 2. I suggest to include figures showing some multifractal spectra either in the main manuscript or as supplementary content.

*Response: We have included a new figure (Fig. 11) as an example showing the joint multifractal spectra between two spatial series of soil water storage measure on 22 October 2008.*

5.- Page 25, Figures 2 and 3. I recommend to show only two or four selected plots of mass exponent functions to increase visibility. (because of thee small size of the Figures, differences are hardly to vew).

*Response: We have modified the figures. Now we have only included only 3 measurement dates for both the figures. The figure numbers have been changed. The new numbers are Fig. 3 and Fig. 4.*

6.- Page 26, Figures 4 and 5. I suggest to take into account the shape of the singularity spectra and not only the amplitude in ths and the Results and in the Discussion sections; also these shapes should provide valuable information, I guess.

*Response: Thank you very much for the comments. We discussed about the shape of the spectra specifically the non-uniformity and the tails of the spectra and their meaning in terms of the distribution of scaling indices (L360-371).*

7.- Page 28, Figure 8. I suggest to move Figure 8 (scheme of the vegetation growth patterns) either to the Material and Method section, or to the section 3.1 (Spatial pattern of soil water storage at different depths). Indeed this Figure is related to the Discussion section, but it is also pertinent to previous section.

*Response: We have moved the figure into materials and methods section and first introduced in L 95. Now the new figure is marked as Fig. 1.*

**Comments from Referees: Anonymous Referee #2**

This work makes use of multifractal analysis and joint multifractal analysis to study the spatio-temporal behavior of soil water storage (SWW) at multiple depths. Several interested implications about the scaling nature of SWW that are relevant to transfer information from one scale to another, are shown. The manuscript is well structured and conclusions are drawn from sound mathematical theories applied to a rich enough database consistent with the algorithms used to estimate theoretical parameters. Therefore, I recommend acceptance following minor revisions.

My specific comments are itemized below: page 6, line 163 and page 6, line 167: I suggest to use Chhabra and Jensen (1989) as a reference instead of Everest and Mandelbrot (1992).

*Response: The reference has been revised from Everest and Mandelbrot (1992) to Chhabra and Jensen (1989) (L187 and L223)*

page 7, line 197 and page 17, line 478: This parameter was first introduced in Caniego, Martín and San José (2003) page 14, line 384: There are two points instead of one to the end of the line. Reference: Caniego, F.J., Martín, M.A. and San José, F., 2003. Rényi dimensions of soil pore size distribution. Geoderma 112 (2003) 205– 216.

*Response: Thanks. We have included the reference (L505).*

[revised manuscript text omitted]

**Commented [r4l8]:** Anonymous #1: recommend to briefly describe the methods used to measure soil water content and to evaluate soil water storage (SWS), even if they have been already detailed described before.

**Commented [r4l9R8]:** Has been added with the contents from Biswas et al., 2012a)

**Commented [r4l10]:** Indicate down to what depth

[revised manuscript text omitted]

Commented [r4I25]: Looks like materials and methods and quite repetitive

Commented [ABP26R25]: It is part of the results and necessary to keep the flow

To sum up, both the unity slope of the $\tau(q)$ curves (Fig. 3 and Fig. 4) and the degree of convexity of the $f(q)$ spectrum (Fig, 6 & Fig. 7) jointly demonstrated that dynamic behavior of surface soil layers in the wet period made SWS highly variable and exhibited multifractal nature, while less environmental forces and increased buffering capacity of deep layers led to monofractal nature. As a result, multiple scaling exponents were required to characterize the variability of SWS in the surface layer during the wet period, while less number of exponents was necessary for deeper layers during wet period or all layers during dry period.

Commented [r4I27]: Is this really need? Repeat the former para

Commented [r4I28R27]: I think this summary is required to increase the readability

[revised manuscript text omitted]

Commented [r4l33]: Would be preferable to show the differential results rather than cumulative plots, which are very smooth and give no indication of multifractality

Fig. 9. Generalized dimension spectra of soil water storage spatial series measured at each 20

cm soil layer down to 140 cm in 2008 for a range of q (-15 to

15 at 0.5 increments).

Fig. 10. Generalized dimension spectra of soil water storage spatial series from surface to different soil layers (cumulative storage) at 20 cm increment down to 140 cm in

2008 for a range of q (-15 to 15 at 0.5 increments).

Fig. 11. Join multifractal spectra between surface (0-20 cm) and immediate (20-40 cm)

subsurface soil layer soil water storage measured on 22 October 2008.

Fig. 11: Conceptual schematics showing vegetation development over time, dominant water loss processes and the scaling behavior of soil water storage at different depths. The figure is developed based on field observations and scaling analysis. The scale of the figure is arbitrary.

**Tables**

Table 1

Commented [r4l34]: Data from 2011 are also shown, please include also 2011 in the caption

[revised manuscript text omitted]

**Commented [r4l36]:** Please indicate the number of data used for each correlation, was it the same for all the dates and depths?

**Commented [r4l37]:** Anonymous Referee #1: I suggest to include figures showing some multifractal spectra either in the main manuscript or as supplementary content.

**Figures**

[Figure]

**Figure 1**

[Figure]

[Figure]

[Figure]

Figure 12

Commented [r4l38]: Please show Y-scale in all left graphs

Commented [r4l39]: Anonymous Referee #1: I recommend to show only two or four selected plots of mass exponent functions to increase visibility. (because of thee small size of the Figures, differences are hardly to view).

[Figure]

[Figure]

Figure 2̶3

Commented [r4l40]: Very difficult to distinguish the points from each depth

Commented [r4l41R40]:

[Figure]

[Figure]

Figure 3̶4̲

**Commented [r4l42]:** Very difficult to distinguish the points from each depth, UM models is missing from the graphs

[Figure]

Figure 5

[Figure]

Figure 46

Commented [r4l43]: Anonymous Referee #1: I suggest to take into account the
shape of the singularity spectra and not only the amplitude in ths and the Results and
in the Discussion sections

[Figure]

Figure 7.

Commented [r4l44]: Anonymous Referee #1: I suggest to take into account the
shape of the singularity spectra and not only the amplitude in ths and the Results and in the Discussion sections, also these shapes should provide valuable information, I guess.

[Figure]

Figure 8

[Figure]

[Figure]

Figure 6̶9

Commented [r4l45]: Some values are overlapped in the Y-axis

[Figure]

[Figure]

Figure 7̶10

[Figure]

Figure 11

[Figure]

Figure 8

Commented [r4l47]: Anonymous Referee #1: I suggest to move Figure 8 (scheme of the vegetation growth patterns) either to the Material and Method section, or to the section 3.1 (Spatial pattern of soil water storage at different depths). Indeed this Figure is related to the Discussion section, but it is also pertinent to previous section.

Commented [r4l48R47]: Once confirmed, I can do the revision.

[Figure]

Figure 912

Commented [r4l49]: The reviewer has asked for three times whether this figure is needed. I have answered in the text, however, maybe need to clarify again in the mail.

---

## Editor Decision (ED1)

**The revised version of the manuscript with reference NPG-2015-81-R1 and entitled "Fractal behaviour of soil water storage at multiple depths" authored by W. Ji, M. Lin, A. Biswas, B.C. Si, H.W. Chau, and H.P. Cresswell and submitted to the Special Issue "Multifractal analysis in soil systems" to be published in Nonlinear Processes in Geophysics represents a great improvement from the former version submitted to the journal. Authors have addressed the comments and suggestions made by the reviewers.**

**However, there are still several minor issues to be corrected prior to an eventual publication in the journal. Pleased, check the following pages for further specifications.**

**Therefore, I still advice for a minor revision prior to the acceptance of the manuscript.**

**Specific comments to the authors:**

*Abstract:*

*Line 16: "deep layers" instead of "deep layer".*

*Line 17: "The current study" instead of "Current study".*

*Lines 24-26: "The dynamic nature of...", this statement is implied in the former two sentences, would you consider removing it, please?*

*Introduction:*

*Line 46: "other than that of measurement" instead of "other than the scale of measurement".*

*Line 47: Remove "scale" after "pedon".*

*Line 48: Remove "scale" after "large catchment".*

*Line 55: "of the scaling process".*

*Line 66: "has" instead of "have".*

*Lines 72-75: "The scaling properties of surface...", please, consider removing this sentence. If you decide to keep it, please, remove the word "characteristics" after "the same" in line 74.*

*Line 84: I would use "multifractal approach" instead of "multifractal analysis".*

*Materials and Methods:*

*Line 91: "differently-sized" instead of "differently sized".*

*Line 94: "late summer" instead of "later summer".*

*Line 96: "Variable water" instead of "Variables water".*

*Line 107: "while deeper layers down to 140 cm were measured" instead of "while the rest deeper soil down to 140 cm depth was measured".*

*Lines 109-110: "Soil water content data was then multiplied by" instead of "These measured data of soil water content from either the neutron probe or TDR were then multiplied with".*

*Line 117: I think it would be useful to add a couple of citations here, at the end of this sentence.*

*Lines 197-198: "One of the widely used...", please, consider re-writing this sentence to "The generalized dimensions were calculated as".*

*Line 207: Remove "the" before "$D_1$" and "$D_0$".*

*Line 225: "was" instead of "is".*

*Line 240: "represent" instead of "represents".*

*Line 244: It should be "a contour plot" or "a contour map" instead of just "a contour".*

*Line 249: Please, check this citation, there is no "Biswas and Si, 2012b" in the reference list.*

*Results:*

*Are units for soil water storage OK? I mean, usually this variable is given in mm and not in cm.*

*Line 252: "the five year period" instead of "five year period".*

*Line 260: "for the surface layer" instead of "for surface".*

*Lines 261-265: This is not clear. Do you mean increases and decreases over time or in depth?*

*Line 269: I would use "that" instead of "and".*

*Line 270: Include "at" before "the deepest layer".*

*Lines 275-276: "A similar trend was also observed for the minimum SWS at different layers". I would remove this sentence since it is already said in the former one.*

*Line 296: "The variability also gradually increased with depth". Sure? Looking at the table you indicate (Supplementary Table S.3) it seems that variability decreased with depth.*

*Line 303: "of three selected dates" instead of "of selected three dates".*

*Line 305: I do not see what you mean by "SWS trend".*

*Line 313: I think that "(single fit)" should be without parenthesis.*

*Lines 321-322: These values are not reported within the supplementary table S.4 as you mentioned*

*here.*

*Line 327: Remove "of soil layers".*

*Line 358: Remove "statistically".*

*Lines 358-359: I do not understand why you referred table S.7 in here.*

*Line 363: "with depth" instead of "with depths".*

*Line 375: Remove "of measurements".*

*Line 393: "years" instead of "year".*

*Line 395: "at all depth layers" instead of "at all layers of cumulative depths".*

*Line 397: Remove "only varied at 3 decimal points".*

*Lines 398-399: Check the subscripts for $D_1$.*

*Lines 409-410: "was also observed at all depth layers" instead of "were also observed at all layers of cumulative depths".*

*Line 415: "demonstrate" instead of "demonstrates".*

*Line 417: "those layers" instead of "the layers".*

*Discussion:*

*Line 442: "factors" instead of "factor".*

*Line 473: "Biswas and Si, 2012", there are a couple of them in the reference list, which one are you referring to?*

*Line 484: Remove "different".*

*Line 487: "values" instead of "value".*

*Line 509: "exhibit a longer" instead of "exhibit longer".*

*Line 519: "from the correlation" instead of "from correlation".*

*Line 535: "and showed stronger similarity to the surface layers", I would remove this.*

*Line 537: "due to the dynamic nature" instead of "due to its dynamic nature".*

*Lines 541-542: I would remove "with less effect from environment factors".*

*Summary and Conclusions:*

*I am not sure that this section is needed since it is basically a repetition of the results.*

*Line 553: "depth" instead of "depths".*

*Line 560: "those of the deep layers" instead of "that of deep layers".*

*References:*

*Lines 583-584: This should be 2012a.*

*Lines 585-587: Since there is no other Biswas et al. 2012, you should remove b after 2012.*

*Lines 588-590: This should be 2012b.*

*Lines 607-609: Why the title of this reference is written in capital letters?*

*Line 636: The "s" should be capital? "Montero, E.S."?*

*Lines 648-650: Why the title of this reference is written in capital letters?*

*Figure captions:*

*Figure 1: I would say "over the landscape" instead of "in the different section of landscapes".*

*Figure 11: This should be the caption for figure 12. In fact, there is no caption that corresponds to figure 11. Please, provide it.*

*Table 1: Please, consider putting "cm" between parentheses in the title of the table, after "soil water storage" and remove it from the columns "average", "maximum", and "minimum".*

*Table 2: Apart from indicating that the number of data points were the same for all the analyses, you could indicate this number, please.*

---

## Author Response (AR2)

Dear Dr. Mirás-Avalos,

Thank you very much for your comments and edits on our manuscript. This really helped a lot modify the paper. I have corrected the suggested edits and some edits as I go through the manuscript again. I have also replied to your comments below.

Hope our submission will be granted.

Thank you very much.

Asim biswas (on behalf of co-authors)

**Comments from the Editor:**

Non-public comments to the Author:

The revised version of the manuscript with reference NPG-2015-81-R1 and entitled "Fractal behaviour of soil water storage at multiple depths" authored by W. Ji, M. Lin, A. Biswas, B.C. Si, H.W. Chau, and H.P. Cresswell and submitted to the Special Issue "Multifractal analysis in soil systems" to be published in Nonlinear Processes in Geophysics represents a great improvement from the former version submitted to the journal. Authors have addressed the comments and suggestions made by the reviewers.

However, there are still several minor issues to be corrected prior to an eventual publication in the journal. Pleased, check the following pages for further specifications.

Therefore, I still advice for a minor revision prior to the acceptance of the manuscript.

*Response: Thank you very much. We have completed the edits in the revised manuscript.*

Specific comments to the authors:

Abstract:

Line 16: "deep layers" instead of "deep layer".

*Response: Corrected*

Line 17: "The current study" instead of "Current study".

*Response: Corrected*

Lines 24-26: "The dynamic nature of…", this statement is implied in the former two sentences, would you consider removing it, please?

*Response: Deleted*

Introduction:

Line 46: "other than that of measurement" instead of "other than the scale of measurement".

*Response: Corrected*

Line 47: Remove "scale" after "pedon".

*Response: Corrected*

Line 48: Remove "scale" after "large catchment".

*Response: Corrected*

Line 55: "of the scaling process".

*Response: Corrected*

Line 66: "has" instead of "have".

*Response: Corrected*

Lines 72-75: "The scaling properties of surface…", please, consider removing this sentence. If you decide to keep it, please, remove the word "characteristics" after "the same" in line 74.

*Response: We kept the sentence but deleted the word.*

Line 84: I would use "multifractal approach" instead of "multifractal analysis".

*Response: We have also used approach rather analysis.*

Materials and Methods:

Line 91: "differently-sized" instead of "differently sized".

*Response: Corrected*

Line 94: "late summer" instead of "later summer".

*Response: Corrected*

Line 96: "Variable water" instead of "Variables water".

*Response: Corrected*

Line 107: "while deeper layers down to 140 cm were measured" instead of "while the rest deeper soil down to 140 cm depth was measured".

*Response: Corrected*

Lines 109-110: "Soil water content data was then multiplied by" instead of "These measured data of soil water content from either the neutron probe or TDR were then multiplied with".

*Response: Modified*

Line 117: I think it would be useful to add a couple of citations here, at the end of this sentence.

*Response: Citations are added*

Lines 197-198: "One of the widely used…", please, consider re-writing this sentence to "The generalized dimensions were calculated as".

*Response: The sentence is rewritten and separated into two smaller sentences*

Line 207: Remove "the" before "D1" and "D0".

*Response: Corrected*

Line 225: "was" instead of "is".

*Response: Corrected*

Line 240: "represent" instead of "represents".

*Response: Corrected*

Line 244: It should be "a contour plot" or "a contour map" instead of just "a contour".

*Response: We used contour plot*

Line 249: Please, check this citation, there is no "Biswas and Si, 2012b" in the reference list.

*Response: We have added the reference in the list.*

Results:

Are units for soil water storage OK? I mean, usually this variable is given in mm and not in cm.

*Response: Yes, you are right as often cases the unit used is mm. However, use of cm in presenting soil water storage is also very common when the storage is in higher amount.*

Line 252: "the five year period" instead of "five year period".

*Response: Corrected*

Line 260: "for the surface layer" instead of "for surface".

*Response: Corrected*

Lines 261-265: This is not clear. Do you mean increases and decreases over time or in depth?

*Response: We have modified the sentence. It is increase with depth*

Line 269: I would use "that" instead of "and".

*Response: Corrected*

Line 270: Include "at" before "the deepest layer".

*Response: Added*

Lines 275-276: "A similar trend was also observed for the minimum SWS at different layers". I would remove this sentence since it is already said in the former one.

*Response: Deleted*

Line 296: "The variability also gradually increased with depth". Sure? Looking at the table you indicate (Supplementary Table S.3) it seems that variability decreased with depth.

*Response: Yes, that was a mistake. We have corrected that.*

Line 303: "of three selected dates" instead of "of selected three dates".

*Response: Corrected*

Line 305: I do not see what you mean by "SWS trend".

*Response: We have modified the sentence. It is not the SWS trend but the trend of scale invariance.*

Line 313: I think that "(single fit)" should be without parenthesis.

*Response: Yes, corrected*

Lines 321-322: These values are not reported within the supplementary table S.4 as you mentioned here.

*Response: table citation deleted*

Line 327: Remove "of soil layers".

*Response: Removed*

Line 358: Remove "statistically".

*Response: Removed*

Lines 358-359: I do not understand why you referred table S.7 in here.

*Response: Citation removed*

Line 363: "with depth" instead of "with depths".

*Response: Corrected*

Line 375: Remove "of measurements".

*Response: Removed*

Line 393: "years" instead of "year".

*Response: Corrected*

Line 395: "at all depth layers" instead of "at all layers of cumulative depths".

*Response: Corrected*

Line 397: Remove "only varied at 3 decimal points".

*Response: Removed*

Lines 398-399: Check the subscripts for D1.

*Response: Corrected*

Lines 409-410: "was also observed at all depth layers" instead of "were also observed at all layers of cumulative depths".

*Response: Corrected*

Line 415: "demonstrate" instead of "demonstrates".

*Response: Corrected*

Line 417: "those layers" instead of "the layers".

*Response: Corrected*

Discussion:

Line 442: "factors" instead of "factor".

*Response: Corrected*

Line 473: "Biswas and Si, 2012", there are a couple of them in the reference list, which one are you referring to?

*Response: We have corrected this. There is only Biswas and Si 2012. Rest are Biswas et al. 2012 (a, b, c).*

Line 484: Remove "different".

*Response: Removed*

Line 487: "values" instead of "value".

*Response: Corrected*

Line 509: "exhibit a longer" instead of "exhibit longer".

*Response: Corrected*

Line 519: "from the correlation" instead of "from correlation".

*Response: Corrected*

Line 535: "and showed stronger similarity to the surface layers", I would remove this.

*Response: Removed*

Line 537: "due to the dynamic nature" instead of "due to its dynamic nature".

*Response: Corrected*

Lines 541-542: I would remove "with less effect from environment factors".

*Response: Removed*

Summary and Conclusions:

I am not sure that this section is needed since it is basically a repetition of the results.

*Response: Yes, it summarizes the whole story. Actually that why we say summary and conclusions rather than only conclusions. I think this summarizes the whole paper. So far we kept the summary.*

Line 553: "depth" instead of "depths".

*Response: Corrected*

Line 560: "those of the deep layers" instead of "that of deep layers".

*Response: Corrected*

*Response: Corrected*

Figure captions:

Figure 1: I would say "over the landscape" instead of "in the different section of landscapes".

*Response: Corrected*

Figure 11: This should be the caption for figure 12. In fact, there is no caption that corresponds to figure 11. Please, provide it.

*Response: Sorry for this mistake. We have added the title for Fig. 11 and corrected the previous version.*

Table 1: Please, consider putting "cm" between parentheses in the title of the table, after "soil water storage" and remove it from the columns "average", "maximum", and "minimum".

*Response: Corrected*

Table 2: Apart from indicating that the number of data points were the same for all the analyses, you could indicate this number, please.

*Response: Corrected*

[revised manuscript text omitted]

Figure 2

[Figure]

Figure 3

[Figure]

Figure 4

[Figure]

Figure 5

[Figure]

Figure 6

[Figure]

Figure 7

[Figure]

Figure 8

[Figure]

Figure 9

[Figure]

Figure 10

[Figure]

Figure 11

[Figure]

Figure 12